# The mesoscale organization of syntaxin 1A and SNAP25 is determined by SNARE–SNARE interactions

**Jasmin Mertins**[1], **Jérôme Finke**[1], **Ricarda Sies**[1], **Kerstin M Rink**[2], **Jan Hasenauer**[3,4,5], **Thorsten Lang**[1]*

[1]Departments of Membrane Biochemistry, Life & Medical Sciences (LIMES) Institute, University of Bonn, Bonn, Germany; [2]Heidelberg University Biochemistry Center (BZH), Heidelberg, Germany; [3]Computational Life Sciences, Life & Medical Sciences (LIMES) Institute, University of Bonn, Bonn, Germany; [4]Interdisciplinary Research Unit Mathematics and Life Sciences, University of Bonn, Bonn, Germany; [5]Institute of Computational Biology, Helmholtz Center Munich – German Research Center for Environmental Health, Neuherberg, Germany

*For correspondence:
thorsten.lang@uni-bonn.de

**Competing interest:** The authors declare that no competing interests exist.

**Abstract** SNARE proteins have been described as the effectors of fusion events in the secretory pathway more than two decades ago. The strong interactions between SNARE domains are clearly important in membrane fusion, but it is unclear whether they are involved in any other cellular processes. Here, we analyzed two classical SNARE proteins, syntaxin 1A and SNAP25. Although they are supposed to be engaged in tight complexes, we surprisingly find them largely segregated in the plasma membrane. Syntaxin 1A only occupies a small fraction of the plasma membrane area. Yet, we find it is able to redistribute the far more abundant SNAP25 on the mesoscale by gathering crowds of SNAP25 molecules onto syntaxin clusters in a SNARE-domain-dependent manner. Our data suggest that SNARE domain interactions are not only involved in driving membrane fusion on the nanoscale, but also play an important role in controlling the general organization of proteins on the mesoscale. Further, we propose these mechanisms preserve active syntaxin 1A–SNAP25 complexes at the plasma membrane.

## Editor's evaluation

In this study, the effect of syntaxin on co-localization of SNAP-25 in plasma membranes of PC12 cells is studied. The degree of co-localization increases as the syntaxin concentration is increased (relative to that of SNAP-25), and this increase depends on the two SNARE domains of SNAP-25, in particular on the N-terminal domain. When Munc18 is added to the plasma membrane sheets, little effect is observed on the syntaxin and SNAP-25 co-localization.

## Introduction

Molecular crowding is a basic feature of biological cells. In some cellular regions, the volume occupied by macromolecules even exceeds that taken by solvating water molecules. Theoretical simulations and in vitro studies suggest that molecular crowding influences thermodynamic equilibria, increases macromolecule activity, promotes molecule assembly, diminishes diffusion rates, and limits steric accessibility (*Zhou et al., 2008*). This implies that in crowded environments macromolecules behave no longer like freely diffusing, single-molecular species.

Molecular crowding routinely occurs in cells wherever large supramolecular assemblies form. Already more than 30 years ago, the plasma membrane has been recognized as a crowded cellular space (*Ryan et al., 1988*). It is constituted of scattered protein-rich and protein-poor regions, each covering roughly half of the area (*Lillemeier et al., 2006*; *Saka et al., 2014*). In mitochondrial membranes, electrophoretic protein migration into a single, maximally crowded patch shows that half of the membrane volume is occupied by proteins (*Sowers and Hackenbrock, 1981*). These protein-rich regions consist of closely associated membrane domains/protein clusters and are referred to as multi-protein assemblies. Some, not yet understood organization principles on the mesoscale determines that different protein types prefer different locations within these multi-protein assemblies (*Saka et al., 2014*). However, a higher level of understanding is difficult to achieve, because these cellular structures are not susceptible to crystallographic analysis, and the high protein density limits mapping of the molecule distribution by fluorescence microscopy.

Two proteins organized in such multi-protein assemblies are the classical SNAREs syntaxin 1A and SNAP25. Together with the vesicle-associated SNARE synaptobrevin 2/VAMP they mediate $Ca^{2+}$-triggered vesicle fusion (*Brunger et al., 2019*). To this end, their SNARE domains assemble into a tight bundle during which the opposed membranes are pulled together until fusion (*Stein et al., 2009*), a process that is highly regulated by accessory proteins, such as Munc18.

Syntaxin 1A is a single-span transmembrane protein with no extracellular segment. Intracellularly, a compact folded N-terminal domain is connected via a long unstructured segment to a SNARE domain, that is attached to the transmembrane segment via a short poly-basic linker region (*Rizo and Xu, 2015*; see also Figure 6). Syntaxin 1A concentrates in clusters exhibiting a high packing density (*Sieber et al., 2007*). The number of molecules per syntaxin cluster has been estimated by three groups employing different approaches. First, a modelling approach based on (1) the experimentally determined cluster density, (2) the molecule copy number per cell, and (3) the diffusion rate (*Sieber et al., 2007*). Second, quantification of fluorescence arising from single clusters (*Knowles et al., 2010*), and third, counting of clustered molecules by PALM (*Rickman et al., 2010*). All studies come to the conclusion that a cluster harbours several dozens of molecules that concentrate on a spot smaller than 100 nm in diameter. Moreover, dSTORM imaging shows that the density of molecules decreases from the cluster centre to its periphery (*Bar-On et al., 2012*).

Clustering occurs at least through two mechanisms. On the one hand, ionic interactions between the polybasic linker region and $PIP_2$ sequesters the protein into larger, less-densely packed, and more dynamic clusters (*Merklinger et al., 2017*; *van den Bogaart et al., 2011*). On the other hand, the SNARE domain immobilizes the protein by packing the molecules more densely into smaller clusters (*Merklinger et al., 2017*).

Like syntaxin 1A, SNAP25 is not homogenously distributed, but its clusters are less pronounced, most likely because they are not forming via the above described two mechanisms involving 'lipid-mediated pre-clustering' and 'cluster tightening by protein–protein interactions'. Instead, SNAP25/SNAP23 enrichment in crowds is only dependent on cholesterol (*Chamberlain and Gould, 2002*) and clustering is not further increased by homophilic SNARE-domain interactions (*Halemani et al., 2010*). One study found dense ensembles of 40–50 molecules (*Rickman et al., 2010*), another study suggested that 50–70 SNAP25 molecules accumulate at secretory vesicle docking sites (*Knowles et al., 2010*). Finally, dSTORM imaging suggests that also in SNAP25 clusters, that are larger and more elliptical than syntaxin 1A clusters, the molecule density diminishes in the periphery (*Bar-On et al., 2012*).

In contrast to syntaxin 1A, SNAP25 has a second SNARE domain, but neither a large N-terminal domain nor a transmembrane region. Instead, it is anchored to the cell membrane by palmitoylation of a cysteine-rich segment located within the linker region that connects the SNARE domains (*Greaves et al., 2010*; see also Figure 6). Different from the SNARE domain of syntaxin 1A, the SNARE domains of SNAP25 are not helical in the isolated protein, yet they become helical in SNARE complexes (*Fasshauer et al., 1997*). Taken together, both proteins, even though very different in structure, form crowds of similar copy number.

An interaction between the first SNARE domain of SNAP25 and syntaxin 1A diminishes SNAP25 mobility (*Halemani et al., 2010*), indicating that the two plasmalemmal SNAREs, as expected, associate. However, microscopy finds no perfect overlap between syntaxin 1A and SNAP25 in neuroendocrine cells and neurons (*Lang et al., 2001*; *Nagy et al., 2005*; *Pertsinidis et al., 2013*; *Rickman et al., 2010*; for one exception in chromaffin cells see *Rickman et al., 2004*).

Here, we study the lateral distribution of syntaxin 1A and SNAP25 in the plasma membrane. We find that both proteins, although forming tight complexes in vitro (*Brunger et al., 2019*; *Jahn and Fasshauer, 2012*), are highly segregated in entities, and that these entities organize specifically via SNARE–SNARE interactions. Because syntaxin 1A is able to determine the distribution of an excess of SNAP25, we propose the syntaxin 1A/SNAP25 entities largely preserve their internal architecture and interact only at their peripheries.

## Results

For studying membrane protein distribution, we employ super-resolution STED (stimulated emission depletion) microscopy, a technique dependent on fluorescent labelling of the target under study. In the crowded environment of the cell membrane, the accessibility of binding probes to SNAP25 is limited, such that antibody labelling only imperfectly visualizes the distribution of SNAP25 (*Zilly et al., 2011*). Therefore, genetic labelling by attaching GFP (green fluorescent protein) to its N-terminus is required, which does not affect the protein's function (*Delgado-Martínez et al., 2007*). Due to the rigid structure of GFP and its molecular size being similar to that of SNAP25, we aimed for labelling only about half of all SNAP25 molecules by GFP-SNAP25 expression, as 100% labelling may interfere with SNAP25 crowding. To test how syntaxin clusters affect the SNAP25 distribution, we express additional syntaxin 1A (Stx-full), or a syntaxin 1A deletion variant being less efficient in forming syntaxin clusters (Stx-ΔS) (*Merklinger et al., 2017*).

### Ratio between SNAP25 and syntaxin 1A

In the P12 cells used in this study, the SNAP25 copy number exceeds the one of syntaxin 1A by 12-fold (*Figure 1—figure supplement 1*), a value in fair agreement with a previously reported 14-fold excess in the plasma membrane (*Knowles et al., 2010*). To make sure that also in our co-expression experiments SNAP25 is more abundant than syntaxin 1A, we analyzed by Western Blot the levels of the expressed constructs in relation to endogenous proteins. We find that expression of GFP-SNAP25 alone or together with Stx-full/Stx-ΔS does not change the endogenous levels of SNAP25 or syntaxin 1A (*Figure 1B, E*). Moreover, trafficking of GFP-SNAP25 to the cell membrane is not affected by co-expression of Stx-full/Stx-ΔS (*Figure 1—figure supplement 2*). GFP-SNAP25 expression leads to a ~30–50% increase of total SNAP25 (endogenous SNAP25+ GFP-SNAP25) (*Figure 1C*), and expression of Stx-full more than doubles the total syntaxin 1A level (*Figure 1F*); Stx-ΔS is equally high expressed as Stx-full (*Figure 1F*) (for equal Stx-full and Stx-ΔS trafficking to the plasma membrane, see *Merklinger et al., 2017*). Hence, in whole cells, SNAP25 and syntaxin 1A are elevated by roughly 50% and 150%, respectively, which diminishes the physiological SNAP25:syntaxin 1A ratio from 12- to 7-fold. Assuming like Knowles et al. that all syntaxin 1A but only 80 % of SNAP25 localize to the plasma membrane, specifically in the plasma membrane, the ratio diminishes by 40% (from 10- to 6-fold). This value, however, is an average including non-transfected and overexpressing cells. To find out the ratio in the plasma membrane of transfected cells, cells were unroofed by a brief ultrasound pulse, leaving behind flat, glass-adhered plasma membrane sheets that are stained for total/overexpressed SNAREs. As shown in *Figure 1—figure supplement 3*, at the plasma membrane, SNAP25 and syntaxin 1A are elevated over endogenous levels at most to 4- and 9-fold, respectively, which reduces the SNAP25:syntaxin 1A ratio maximally to 4-fold. In membrane sheets from the middle expression range, the ratio diminishes less, to about 5-fold, showing that SNAP25 still largely exceeds the concentration of syntaxin 1A in our experiments.

### Syntaxin 1A expression diminishes GFP accessibility

In a previous study, the antibody accessibility of SNAP25 was strongly dependent on the conformational state of the protein; antibody staining was reduced by ~90 % after $Ca^{2+}$-induced protein clustering. Moreover, SNAP25 clustering reduced the emission of its attached fluorescent protein, most likely due to self-quenching (*Zilly et al., 2011*), an effect described to occur in GFP-labelled oligomers (*Ochiishi et al., 2016*; *Schneider et al., 2021*). In the following, we test whether elevation of syntaxin 1A has any influence on the conformational state of SNAP25, possibly reducing its antibody accessibility. It should be noted that syntaxin 1A binds to the SNARE domains of SNAP25 and by this may prevent antibody binding to SNAP25 independent from the SNAP25 packing state. Therefore,

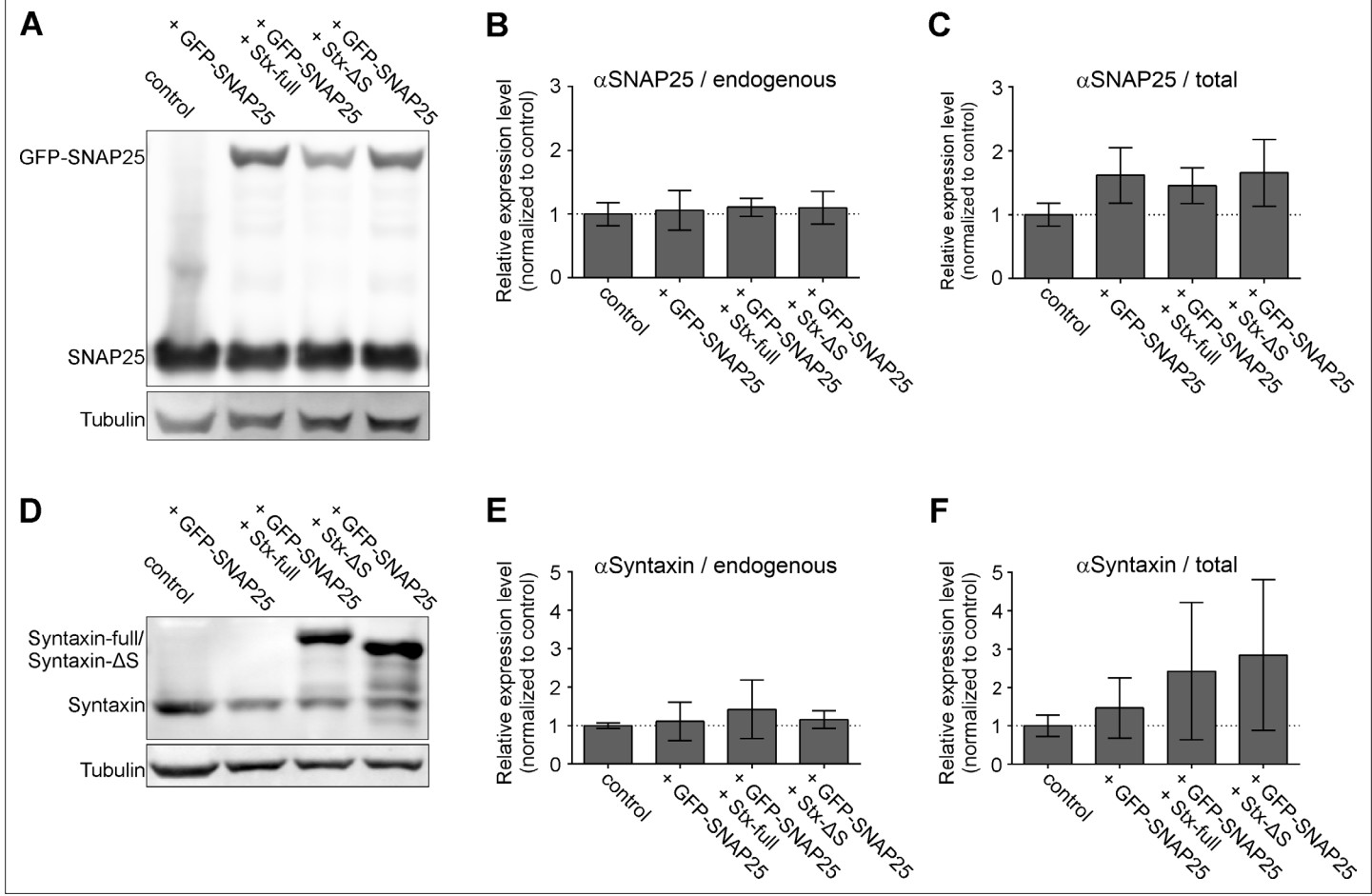

**Figure 1.** Expression levels of endogenous SNAREs and overexpressed GFP-SNAP25, Stx-full, and Stx-ΔS. PC12 cells are transfected with GFP-SNAP25 alone or in combination with Stx-full/Stx-ΔS. The sample is split for parallel analysis of expression levels by Western Blot and molecular accessibility to GFP-SNAP25 (*Figure 2*). (**A**) Western Blot analyzing the expression levels of endogenous SNAP25 (25 kDa) and GFP-SNAP25 (52 kDa) using an antibody raised against the N-terminal SNARE domain of SNAP25. Band intensities are related to the tubulin light chain bands, followed by normalization to control (untransfected PC12 cells). (**B**) Quantification of the 25 kDa band of endogenous SNAP25 and (**C**) total SNAP25 (endogenous SNAP25 and the 52 kDa band of GFP-SNAP25). (**D**) Western Blot analysis of endogenous syntaxin 1A (33 kDa), overexpressed syntaxin 1A (Stx-full; ~38 kDa with a triple myc-tag fused via a linker region to the C-terminus) and Stx-ΔS (~35 kDa; Stx-full lacking the N-terminal half of the SNARE domain) using an antibody detecting the N-terminal domain of syntaxin 1A. (**E**) Normalized band intensities of endogenous syntaxin. (**F**) Signal of endogenous syntaxin (without or with GFP-SNAP25 overexpression) and endogenous syntaxin 1A + Stx-full or Stx-ΔS with GFP-SNAP25 overexpression. Values are given as means ± standard deviation (SD). Statistical analysis showed no significant difference between any of the conditions ($n = 3$ experiments; two-tailed unpaired $t$-test, not significant = $p > 0.05$).

The online version of this article includes the following source data and figure supplement(s) for figure 1:

**Figure supplement 1.** Ratio between endogenous SNAP25 and syntaxin 1A.

**Figure supplement 1—source data 1.** Raw Western Blot images and annotated full blots for *Figure 1—figure supplement 1A*.

**Figure supplement 2.** GFP-SNAP25 targeting to the plasma membrane.

**Figure supplement 3.** Elevation of SNAP25 and syntaxin 1A over endogenous levels in plasma membrane sheets.

**Source data 1.** Raw Western Blot images and annotated full blots for *Figure 1A, D*.

we probed the molecular accessibility by targeting the GFP-tag instead of the SNARE domain using a fluorescent-labelled anti-GFP nanobody. As a fluorophore we selected ATTO647, because in contrast to GFP it is suitable for STED microscopy, allowing us to use the same samples for studying the meso-scale organization at super-resolution (see below).

Accessibility to SNAP25 was analyzed specifically in the plasma membrane employing membrane sheets that are fixed, stained, and imaged by epifluorescence microscopy. High overlap between the

signal arising from GFP and the GFP-bound nanobody indicates that the nanobody reliably reports the overall distribution of GFP-SNAP25 (*Figure 2—figure supplement 1*).

Plotting the nanobody- against GFP-SNAP25 signal from individual membrane sheets (control in *Figure 2*; see black regression line in *Figure 2B*) shows that nanobody binding positively correlates with the GFP-SNAP25 expression level. Expressing additional syntaxin (Stx-full), the slope of the regression line is much smaller (magenta line in *Figure 2B*). That under this condition the same number of GFP molecules correlates with a lower intensity in the ATTO647 channel suggests that syntaxin 1A decreases the accessibility of the nanobody to the GFP-tag of SNAP25. This observation is further supported by a trend towards a smaller Pearson correlation coefficient (PCC) between GFP and ATTO647 that decreases from 0.73 to 0.62 (*Figure 2A* and *Figure 2—figure supplement 2A*). Finally, we also find a diminishment of the average GFP intensity (*Figure 2—figure supplement 2B*), pointing to the possibility of GFP self-quenching. Together, the three analyzed parameters all support the view that an elevation of syntaxin 1A is associated with a change in the packing state of SNAP25.

To test whether reduced nanobody accessibility is caused by a syntaxin-specific function, we expressed instead of Stx-full the construct Stx-ΔS with a shortened SNARE domain that is less efficient in clustering (*Merklinger et al., 2017*). Additionally, its interaction with SNAP25 should be weakened. To obtain an estimate on how much the interaction is affected, we employed co-immunoprecipitation of GFP-SNAP25 and Stx-full/Stx-ΔS (*Figure 3*). For this experiment, we turned to HepG2 cells that express no endogenous syntaxin 1A and SNAP25. GFP-SNAP25 is immunoprecipitated by anti-GFP-coated agarose beads and the pull-down of Stx-full/Stx-ΔS is analyzed by Western Blot. Compared to Stx-full, only 10% of Stx-ΔS is co-immunoprecipitated (*Figure 3*). This suggests that the interaction between Stx-ΔS and SNAP25 is not completely abolished, but strongly reduced.

In accordance with that, in the accessibility assay, Stx-ΔS has a markedly smaller effect on the slope of the regression line (see green line in *Figure 2B*) and does not reduce the PCC (*Figure 2A*, *Figure 2—figure supplement 2A*). Moreover, it does not decrease the GFP intensity (*Figure 2—figure supplement 2B*). In summary, the data show that syntaxin 1A has a much stronger effect on the molecular accessibility of SNAP25 than a variant interacting only weakly with SNAP25.

## Syntaxin 1A clusters reorganize SNAP25 crowds via a SNARE–SNARE interaction

We next analyzed the lateral distribution of SNAP25 and syntaxin 1A on membrane sheets using two-colour super-resolution STED microscopy at ~65 nm resolution (*Figure 4—figure supplement 1*). For syntaxin 1A visualization, antibody labelling is employed. Labelling is sub-stoichiometric because the globular size of antibodies/F$_{ab}$ fragments exceeds the distances between epitopes in a syntaxin 1A cluster (see also Figure 6). However, sub-stoichiometric detection is not problematic as long as the overall distribution of syntaxin 1A is reported correctly. This is the case when the antibody HPC-1 is used. It is raised against the large N-terminal domain of syntaxin 1A (Figure 6) and properly visualizes its distribution (*Zilly et al., 2011*).

In control images, syntaxin 1A concentrates in sharply defined clusters (*Figure 4A*, top left) of ~80 nm in diameter at a density of ~11 clusters per µm² that occupy only a minor fraction of the membrane area.

The distance from one syntaxin 1A cluster to the nearest other syntaxin clusters is different from the one of uniformly randomly distributed clusters behaving like particles in an ideal gas (*Figure 4—figure supplement 2*). Compared to an ideal gas, the most likely distances are larger, peaking around 180 nm. This observation suggests a non-random distribution of endogenous syntaxin 1A clusters. In contrast, SNAP25 is present in diffuse spotty structures with a diameter of ~160 nm that are spread all over the membrane (*Figure 4A*, top middle).

Overexpression of syntaxin 1A (Stx-full) increases to ~5-fold the average/median (543%/514%) syntaxin staining intensity (*Figure 4B*), which is accompanied by a tripling in the number of maxima and an increase in maxima size by 40 % (*Figure 4B*). These figures are in accordance with previous reports characterizing the relationship between syntaxin expression level and clustering behaviour (*Merklinger et al., 2017*; *Sieber et al., 2006*). Finally, with reference to *Figure 1—figure supplement 3*, the above-mentioned 5-fold increase in staining suggests that we imaged membrane sheets from the middle overexpression range, with a still 5-fold excess of SNAP25 over syntaxin 1A.

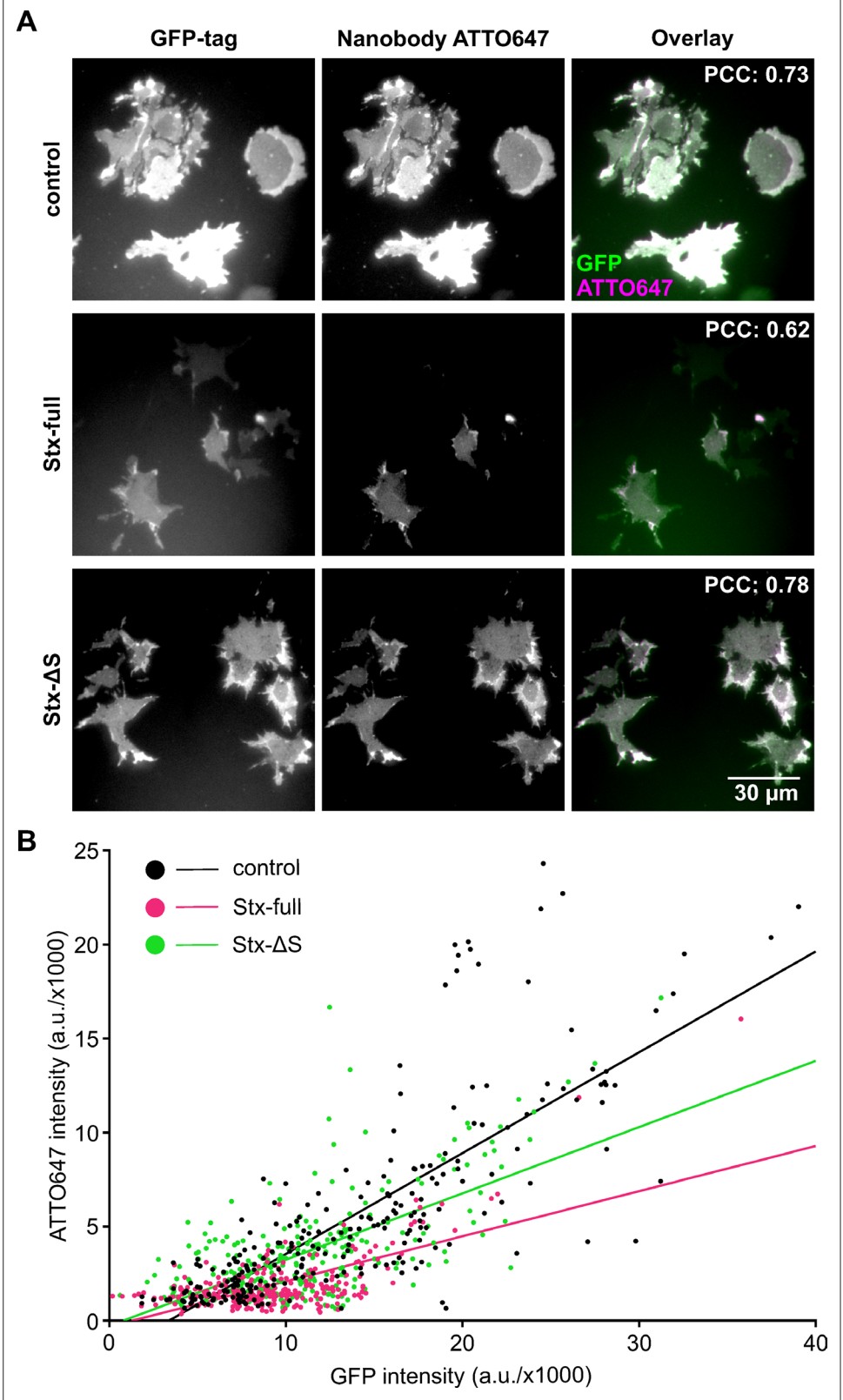

**Figure 2.** Syntaxin 1A overexpression reduces nanobody accessibility to GFP-SNAP25. Cells from the same preparations as used for Western Blot analysis (*Figure 1*) were unroofed, leaving behind glass-adhered plasma membrane sheets that are fixed and stained with an ATTO647-labelled GFP nanobody. (**A**) Epifluorescence micrographs recording GFP- (left) and ATTO647 fluorescence (middle) arising from GFP-SNAP25 and anti-GFP

*Figure 2 continued on next page*

*Figure 2 continued*

nanobody (ATTO647 labelled), respectively. Membrane sheets are generated from cells expressing GFP-SNAP25 alone (control) or together with Stx-full/Stx-ΔS. Images from one channel are shown at the same scaling employing a linear lookup table. The average Pearson correlation coefficient (PCC) between the two channels is stated in the top right corner, for statistics see *Figure 2—figure supplement 2A*. (**B**) From individual membrane sheets, mean ATTO647 intensity is plotted against mean GFP intensity, followed by fitting of a linear regression line (slope values are 0.54 [control], 0.24 [Stx-full], and 0.35 [Stx-ΔS]). Two-tailed *t*-test shows a p value < 0.0001 between control and Stx-full/Stx-ΔS and between Stx-full and Stx-ΔS slopes. Per condition, membrane sheets from three experiments are pooled (control, *n* = 245; Stx-full, *n* = 266; Stx-ΔS, *n* = 269 membrane sheets). For the average GFP signal intensity, see *Figure 2—figure supplement 2B*.

The online version of this article includes the following figure supplement(s) for figure 2:

**Figure supplement 1.** GFP-SNAP25 detection by an anti-GFP nanobody.

**Figure supplement 2.** Pearson correlation coefficients (PCCs) and average GFP fluorescence intensity.

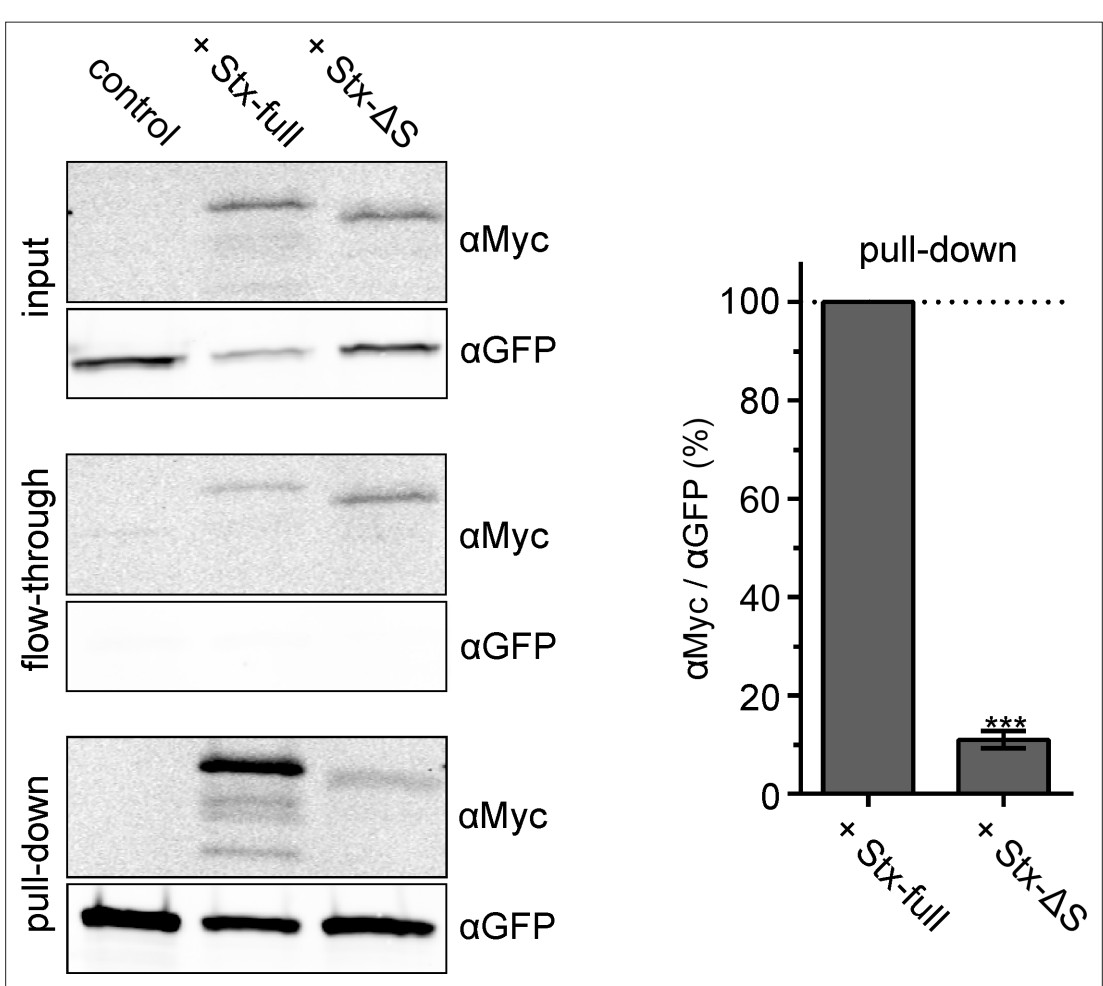

**Figure 3.** Interaction between SNAP25 and Stx-full/Stx-ΔS. HepG2 cells that do not endogenously express syntaxin 1A and SNAP25 express GFP-SNAP25 alone (control) or together with Stx-full/Stx-ΔS. For immunoprecipitation of GFP-SNAP25, cell lysate is incubated with anti-GFP agarose beads. Left, Western Blot detecting Stx-full/Stx-ΔS using an anti-Myc-tag antibody (αMyc) and GFP-SNAP25 using an anti-GFP antibody (αGFP) in the input (cell lysate), flow-through and pull-down. Right, band quantification of pull-down. The αMyc-band intensity is related to the αGFP-band intensity and Stx-full is set to 100 %. Values are given as means ± standard deviation (SD; *n* = 3 experiments). Two-tailed paired *t*-test compares Stx-full to Stx-ΔS; ***p < 0.001.

The online version of this article includes the following figure supplement(s) for figure 3:

**Source data 1.** Raw Western Blot images and annotated full blot for *Figure 3*.

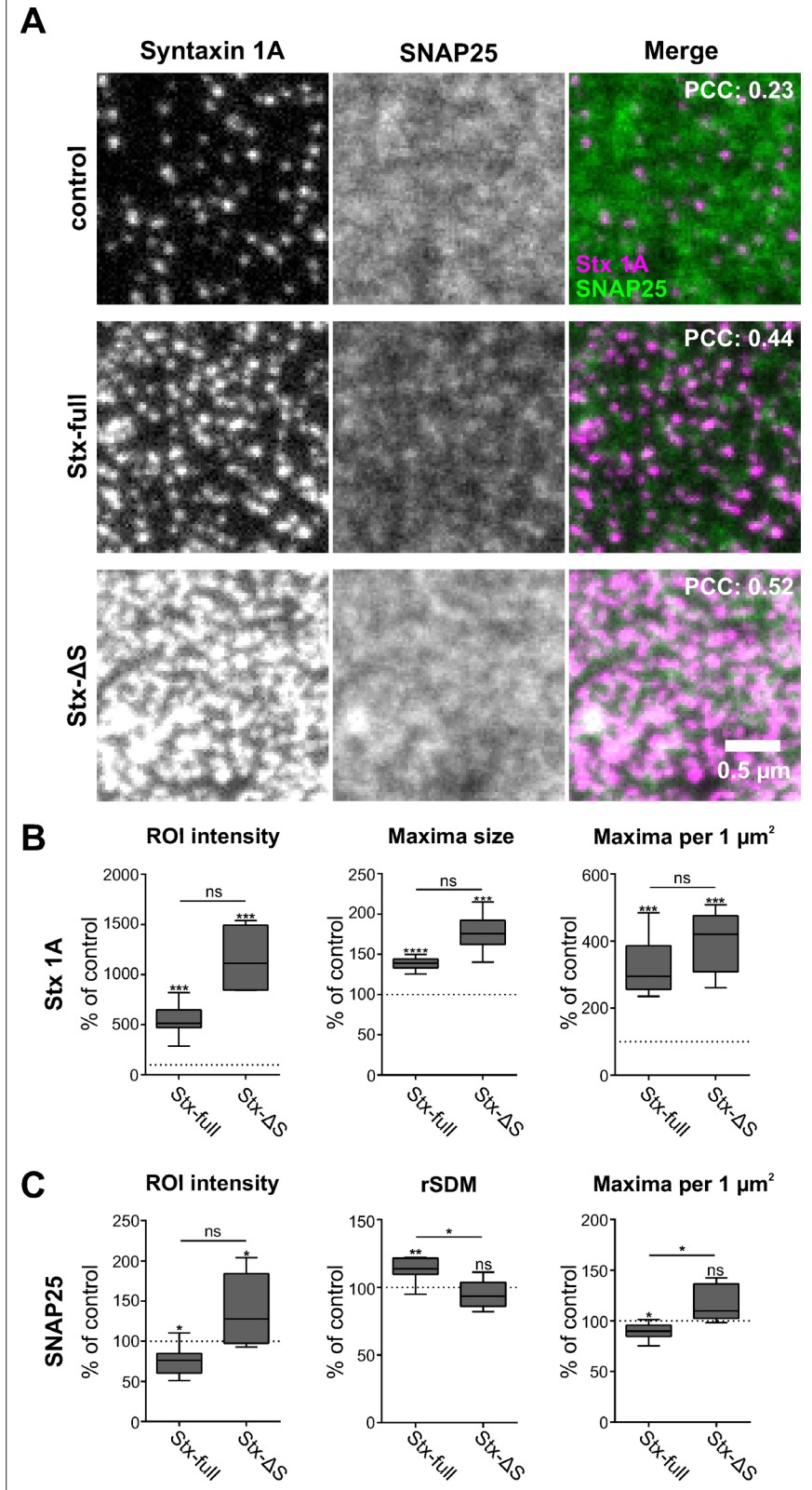

**Figure 4.** Syntaxin clusters change the distribution of SNAP25. Membrane sheets are generated from PC12 cells expressing GFP-SNAP25 alone (control) or together with Stx-full/Stx-ΔS. Endogenous syntaxin/syntaxin constructs are recognized by an antibody raised against the molecule's N-terminus, visualized by an AlexaFluor594-coupled secondary antibody; GFP-SNAP25 is visualized by an ATTO647-labelled GFP nanobody. (**A**) Representative images

*Figure 4 continued on next page*

*Figure 4 continued*

from the syntaxin 1A and SNAP25 channels of the control (only endogenous syntaxin), Stx-full (+ additional syntaxin), and Stx-ΔS (+ additional syntaxin with a shortened SNARE domain) condition. For magnified views see *Figure 4—figure supplement 4*. Top right corner, average Pearson correlation coefficient (PCC) between the two channels; for statistics see *Figure 4—figure supplement 5*. (**B**) Change of the syntaxin 1A staining pattern after Stx-full/Stx-ΔS expression. Staining intensity (region of interest [ROI] intensity), maxima size, and maxima density are related to the average control values (set to 100%). (**C**) SNAP25 staining pattern after Stx-full/Stx-ΔS expression. SNAP25 staining intensity (ROI intensity), clustering degree (relative standard deviation of the mean [rSDM]; 100% = 0.47), and maxima density are related to the control. For the relation between rSDM and Stx-full expression level, see *Figure 4—figure supplement 6*. Values are shown as box plots ($n$ = 6–9 experiments, including three experimental preparations used as well for *Figure 2*; at least 15 membrane sheets were imaged per condition and experiment), showing the median, the 25th and 75th percentile (box), and the minimum and maximum value. Two-tailed paired Student's $t$-test compare (1) control to Stx-full/Stx-ΔS and (2) Stx-full to Stx-ΔS; ns = not significant, $p > 0.05$, *$p < 0.05$, **$p < 0.01$, ***$p < 0.001$. After exchanging the spectral properties of the labels using ATTO647 for the syntaxin and ATTO594 for the SNAP25 channel, the same trends are observed. The syntaxin 1A maxima are sharper and more defined than SNAP25 maxima as well (*Figure 4—figure supplement 7*).

The online version of this article includes the following figure supplement(s) for figure 4:

**Figure supplement 1.** Size of the point spread functions in the AlexaFluor594 and ATTO647N channels.

**Figure supplement 2.** Comparing real syntaxin 1A maxima positions to a uniformly random distribution.

**Figure supplement 3.** Average number of syntaxin 1A maxima at a specific distance around SNAP25 maxima, probability density function, and radial pair distribution function.

**Figure supplement 4.** SNAP25 accumulates at syntaxin clusters.

**Figure supplement 5.** Pearson correlation coefficient (PCC) and corresponding controls.

**Figure supplement 6.** Relative standard deviation of the mean (rSDM) dependence from syntaxin expression level.

**Figure supplement 7.** STED analysis with exchanged spectral properties of the SNAP25 and syntaxin 1A channels.

Expression of Stx-ΔS increases to ~11-fold the average/median (1154%/1112%) staining intensity (*Figure 4B*), more than triples the maxima number, and further increases maxima size. In comparison to Stx-full a stronger staining is expected, because due to better epitope accessibility antibodies stain the molecules in the loosely packed Stx-ΔS clusters 3-fold more efficiently than in Stx-full clusters, as shown in cells lacking endogenous syntaxin 1A and SNAP25 (*Merklinger et al., 2017*). To relate the previously reported 3-fold increase to the increase in *Figure 4*, correction for endogenous syntaxin staining is required. Subtracting 100 % as offset ((1154% − 100% offset)/(543% − 100% offset) = 2.4), in *Figure 4B* Stx-ΔS staining is increased over Stx-full staining by 2.4-fold, which is in the expected 3-fold range (*Merklinger et al., 2017*) when assuming similar expression levels (*Figure 1F*).

Elevation of Stx-full causes a change in the SNAP25 distribution, which becomes less uniformly distributed. The SNAP25 area coverage decreases, with more deserted areas forming between SNAP25-rich regions (*Figure 4A*, middle row, centre image). The visual impression is confirmed by image analysis revealing an increase in the relative standard deviation of the mean (rSDM, *Figure 4C*), which is a parameter describing the degree of clustering (*Zilly et al., 2011*). Moreover, we observe a small decrease in spots per $\mu m^2$ (*Figure 4C*). We assume that this decrease is largely underestimated due to the less defined SNAP25 signal, which leads to spot over-counting. Finally, as already observed in the epitope accessibility assay (*Figure 2*), the nanobody accessibility to GFP is diminished (*Figure 4C*). None of these changes is observed upon expression of Stx-ΔS (*Figure 4C*), indicating that they depend on a fully functional syntaxin 1A SNARE motif.

Overexpression of both Stx-full and Stx-ΔS doubles the PCC between the syntaxin and the SNAP25 channel (*Figure 4A*, see values stated in the overlay images). This is counterintuitive when assuming that only Stx-full has an effect on SNAP25 distribution, but can be explained. Overexpression of Stx-full and Stx-ΔS each strongly increases the number of syntaxin maxima at a specific distance from SNAP25 maxima (see *Figure 4—figure supplement 3A*). In accordance with this we observe substantial shortening of the distance between a SNAP25 and its nearest syntaxin maxima (see *Figure 4—figure supplement 3B*). This results under both conditions in an increase of the overlap between the syntaxin and SNAP25 channels, finally leading to similar increases in the PCC.

Altogether, the data suggest that syntaxin clusters recruit the more widespread distributing SNAP25 crowds via the syntaxin 1A SNARE domain, whereas Stx-ΔS clusters only intermingle with SNAP25 crowds without changing their distribution.

SNAP25 has two SNARE domains that are integral part of the ternary SNARE complex forming during membrane fusion (*Stein et al., 2009*). Next, we ask whether only one or both SNARE domain(s) are required for mesoscale organization. To this end, we compare SNAP25 to three deletion constructs, deleting either the N-terminal SNARE domain (SNAP25-ΔN), the C-terminal SNARE domain (SNAP25-ΔC), or both domains (SNAP25-ΔN/C) (see cartoons in *Figure 5A*), that are expressed together with Stx-full (the condition 'SNAP25-WT' in *Figure 5* is identical to the condition 'Stx-full' in *Figure 4*).

As illustrated by the images in *Figure 5A*, the mean signal intensity of SNAP25-wildtype (SNAP25-WT) was the lowest of all SNAP25 constructs. Moreover, SNAP25-WT showed the highest PCC with the syntaxin channel (*Figure 5B*, *Figure 5—figure supplement 1*). A significant decrease in the PCC and the rSDM is observed upon deletion of the N-terminal SNARE domain in the constructs SNAP25-ΔN and SNAP25-ΔN/C (*Figure 5B, C*). It should be noted that the rSDM decreases with higher mean intensity, raising the possibility that the rSDM of SNAP25-WT is only larger because its signal intensity is lower. However, this is not the case, because when similar intensities are compared on the level of individual membrane sheets, the rSDM of SNAP25-WT is larger than the one of SNAP25-ΔN/C (*Figure 5D*, see magnified view). Analyzing the number of syntaxin 1A maxima at a specific distance from SNAP25 maxima and the probability density function analysis undermines the PCC and rSDM results. Smallest distances between maxima in the syntaxin and SNAP25 channels are most frequent when SNAP25-WT is expressed, followed by SNAP25-ΔC, and SNAP25-ΔN/SNAP25-ΔN/C (*Figure 5—figure supplement 2*). Hence, the N-terminal SNARE-domain of SNAP25 is required for recruitment to syntaxin clusters. Higher fluorescence intensities of SNAP25-ΔN and SNAP25-ΔN/C (see *Figure 5A*) can be explained by better molecular accessibility to the nanobody because of less/no recruitment to syntaxin clusters.

We also wondered whether the SNARE domains are sufficient for mesoscale organization, or if perhaps Munc18 influences mesoscale organization by controlling the syntaxin 1A/SNAP25 interaction (see e.g. *Burkhardt et al., 2008*; *Dulubova et al., 1999*; *Ma et al., 2013*; *Rickman et al., 2007*; *Weninger et al., 2008*; *Zilly et al., 2006*). Membrane sheets from cells expressing GFP-SNAP25 were incubated for 15 min with control buffer or 6 µM Munc18, fixed, stained, and analyzed by STED microscopy (*Figure 5—figure supplement 3*). As previously reported, incubation of membrane sheets causes syntaxin 1A patching, resulting in a more clustered pattern (*Zilly et al., 2011*). This is noted in a trend towards a lower number of maxima (*Figure 5—figure supplement 3B*; see also *Bar-On et al., 2012*). On the other hand, SNAP25 maxima appear unchanged (*Figure 5—figure supplement 3C*). Hence, addition of Munc18 has no significant effect on the rSDM and the maxima per µm$^2$ and consequently appears to have no influence on the SNAP25 mesoscale distribution, although it may have one on the nanoscale organization (e.g. the formation of complexes between single syntaxin 1A and Munc18 proteins).

## Discussion
### Visualization of proteins by fluorescent labelling and the interpretation of staining patterns

For studying the subcellular distribution of biomolecules at nanoscale resolution, specific fluorescent labelling is inevitable. Under ideal conditions, each copy of a specific protein is labelled by a single fluorophore, and each fluorophore is detected with equal efficiency. However, this cannot be realized in tightly packed crowds in which the epitope for labelling is obscured and not accessible to binding probes. Turning to GFP, genetic labelling allows for 100 % labelling efficiency. However, apart from the possibility that GFP may affect molecule packing, self-quenching of close (e.g. less than 5 nm) fluorophores results in signal underestimation (*Ochiishi et al., 2016*; *Schneider et al., 2021*). Consequently, not all molecules in a tightly packed crowd are detected.

In our experiments, we observe different signal intensities at the same molecule number. Regarding GFP-SNAP25, the ratio between nanobody and GFP intensity is reduced upon overexpression of syntaxin 1A (*Figure 2*). We argue that the change in nanobody intensity is caused by differences in

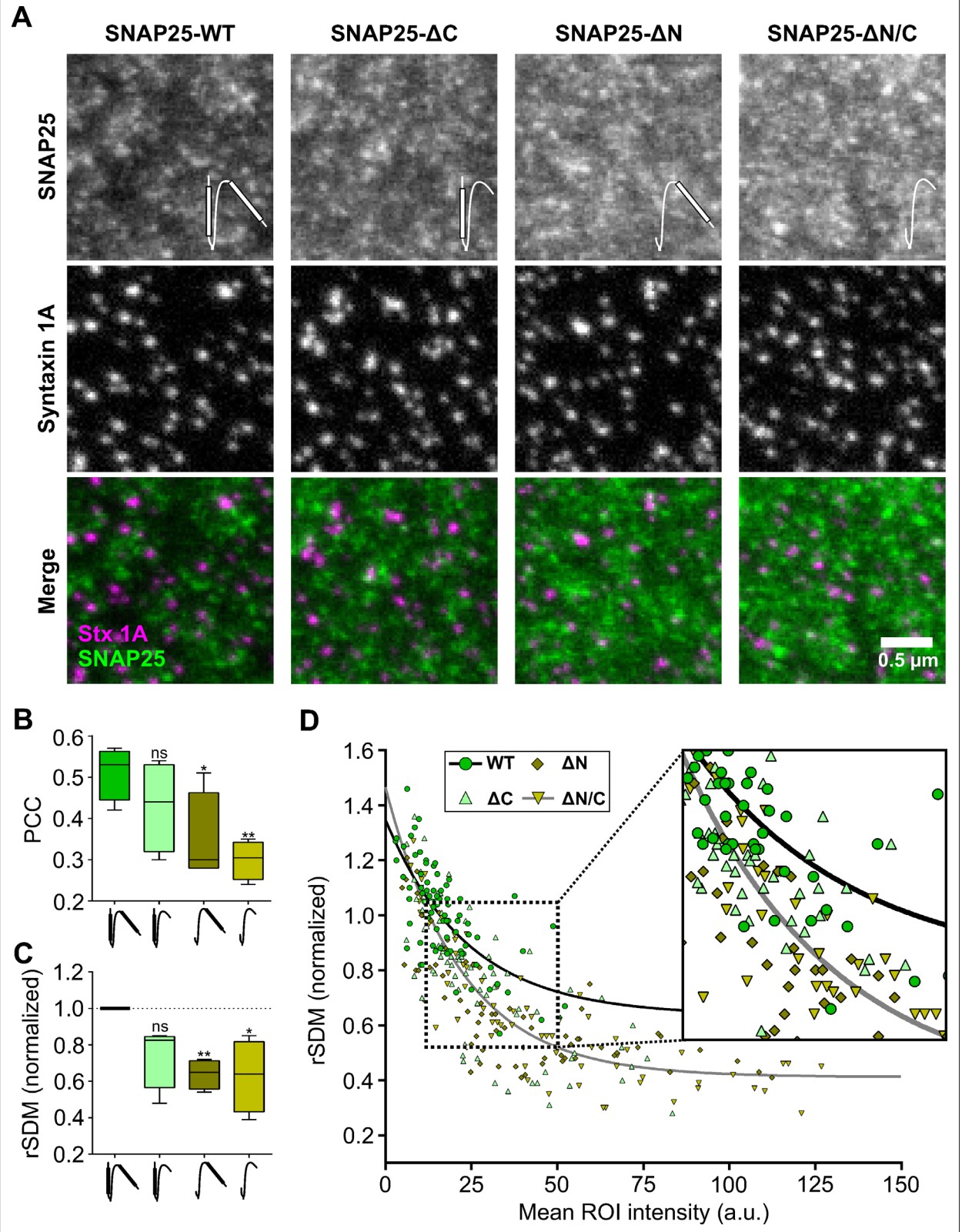

**Figure 5.** SNAP25 crowds bind to syntaxin 1A clusters via the N-terminal SNARE domain of SNAP25. Stx-full (syntaxin 1A is stained by the HPC-1 antibody visualized by an AlexaFluor594-coupled secondary antibody) is overexpressed in PC12 cells together with GFP-SNAP25 or GFP-SNAP25 lacking either the C-terminal SNARE domain (SNAP25-ΔC), the N-terminal SNARE domain (SNAP25-ΔN), or both (SNAP25-ΔN/C) (visualized by an anti-GFP nanobody coupled to ATTO647). Membrane sheets are generated and analyzed exactly as in *Figure 4*. (**A**) Representative images from the

*Figure 5 continued on next page*

*Figure 5 continued*

different channels and overlays. Please note that SNAP25-wildtype (SNAP25-WT) staining was on average 40 % dimmer than for example SNAP25-ΔN. Pictograms in the upper image row illustrate the SNARE domains of SNAP25 that are depicted as white bars. (**B**) Pearson correlation coefficient (PCC) between GFP and syntaxin 1A channel; for control see *Figure 5—figure supplement 1*. (**C**) Clustering degrees normalized to SNAP25-WT (100 % = 0.523). Values are shown as box plots (n = 4 experiments; at least 10 membrane sheets were imaged per condition and experiment), showing the median, the 25th and 75th percentile (box), and the minimum and maximum value. Two-tailed Student's *t*-test compare SNAP25-WT to SNAP25-ΔC/ SNAP25-ΔN/SNAP25ΔN/C; ns = not significant, p > 0.05, *p < 0.05, **p < 0.01. (**D**) From individual membrane sheets, the degree of clustering (relative standard deviation of the mean, rSDM) is plotted against the mean region of interest (ROI) intensity of the GFP staining. Black and grey lines illustrate fits of a non-linear regression line ($y = (a − b)e^{−d*x} + b$) to the values of SNAP25-WT (a = 1.35, b = 0.64, d = 0.042) and SNAP25-ΔN/C (a = 1.47, b = 0.41, d = 0.046), respectively.

The online version of this article includes the following figure supplement(s) for figure 5:

**Figure supplement 1.** Pearson correlation coefficient (PCC) controls.

**Figure supplement 2.** Average number of syntaxin 1A maxima at a specific distance around SNAP25/SNAP25 deletion constructs maxima, probability density function, and radial pair distribution function.

**Figure supplement 3.** Munc18-1 does not influence SNAP25 mesoscale organization.

GFP accessibility that most likely decreases due to shielding by syntaxin 1A's N-terminal domain (see *Figure 6*). Yet, we cannot rule out a component of ATTO647-fluorophore quenching if two nanobodies get very close. In addition, we may underestimate GFP intensity due to self-quenching (see trend towards lower GFP intensity upon overexpression of syntaxin 1A in *Figure 2—figure supplement 2B*). On the other hand, Stx-ΔS labelling yields a larger staining intensity than Stx-full labelling (*Figure 4B*; see also *Merklinger et al., 2017*). Hence, we deal for each protein with a unique relationship between packing state/conformational change and signal intensity. However, the lateral changes in the SNAP25 distribution cannot be explained by distortions in the visualization of proteins and are due to syntaxin 1A-mediated changes in its organization. Therefore, we can safely conclude that upon overexpression of syntaxin 1A the signal of SNAP25 becomes dimmer and more clustered because SNAP25 crowds are recruited to syntaxin clusters via a SNARE–SNARE interaction.

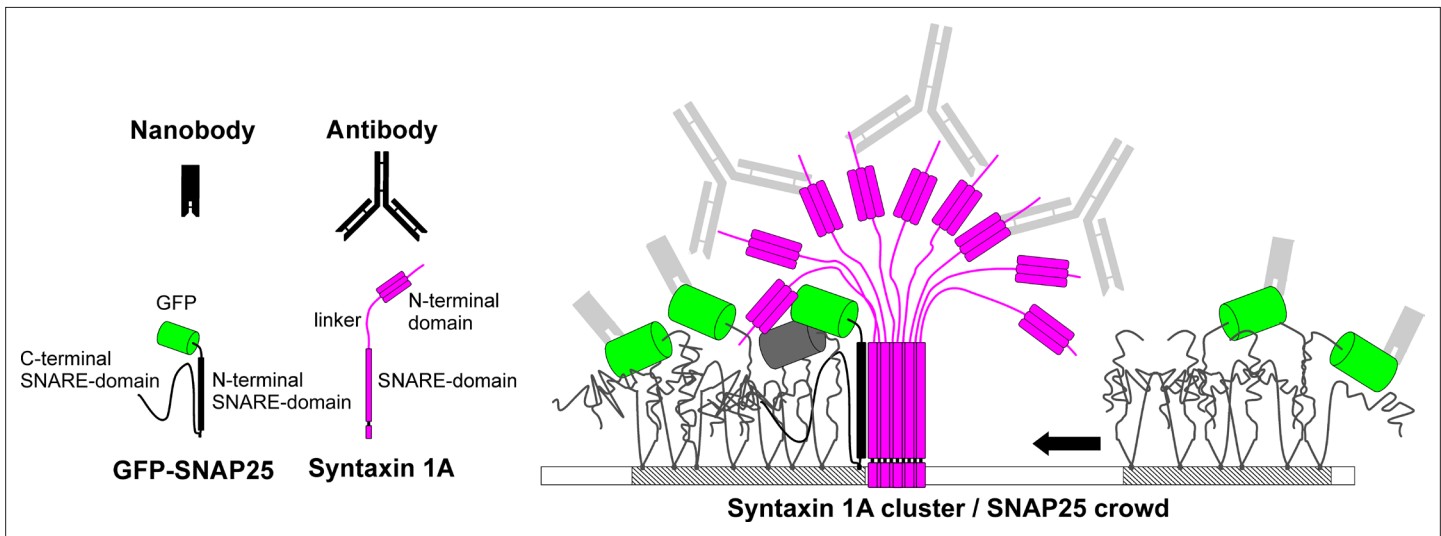

**Figure 6.** Model of SNAP25 crowd recruitment to a syntaxin 1A cluster. Syntaxin 1A molecules form tightly packed clusters resembling a bunch of flowers. SNAP25 is enriched in cholesterol-dependent lipid phases (shaded membrane). In the unbound crowd (right), the GFP-tags are able to move freely. Hence, it is less likely that two GFPs are close enough for self-quenching. Moreover, in the unbound crowd all GFPs are accessible to the nanobody. This is different in the bound crowd (left). Here, the N-terminal SNARE domain of SNAP25 interacts with the surface of the syntaxin 1A cluster, positioning the crowd close to the trunk such that some of the GFP-tags are shielded under the cluster's roof and are no longer accessible to nanobody binding. The GFPs pack more densely and are immobilized, which promotes self-quenching (see grey GFP barrel). One syntaxin 1A cluster can recruit several SNAP25 crowds, explaining how syntaxin controls a surplus of SNAP25. The legend shows the binding probes, GFP-SNAP25 and syntaxin 1A. SNARE domains of SNAP25 are unstructured, unless they participate in an interaction with syntaxin 1 A during which they become alpha-helical (in the cartoon alpha-helicity is indicated by drawing the SNARE domain as a rectangle).

## Mesoscale organization of the plasma membrane

For decades, the in-homogenous distribution of proteins and lipids in cellular membranes is under study. Many studies describe the presence of various membrane proteins being organized in structures referred to as rafts, membrane microdomains, or clusters (*Destainville et al., 2016*; *Lingwood and Simons, 2010*). When comparing the distribution of 'all proteins' to 'a specific protein' one finds that proteins are either in protein-rich or depleted areas, and within the protein-rich area, a specific protein forms its own cluster at a preferential position, known as the mesoscale organization of the plasma membrane (*Saka et al., 2014*).

We find that the mesoscale organization can be explained by specific interactions between protein crowds. In the case of syntaxin 1A and SNAP25, each protein tends to form its own crowd. While SNAP25 crowds are more diffuse in appearance, syntaxin 1A forms tighter crowds (clusters) that recruit SNAP25 crowds via the SNARE domain, and by this, SNAP25 becomes less uniformly distributed, as noticed in a larger rSDM (*Figure 4C*). We estimate that in our experiments the physiological SNAP25:syntaxin 1A ratio is roughly halved from 10- to 5-fold. However, the rSDM increase is not related to the diminishment of the physiological ratio, or an increase in SNARE concentration, because as shown in *Figure 4—figure supplement 6*, we do not observe a trend towards larger rSDMs upon stronger overexpression.

Using SNAP25 deletion constructs (*Figure 5*), we further specify that for recruitment the N-terminal SNARE domain of SNAP25 is relevant, whereas deletion of the C-terminal domain has no significant effect, although trends towards a smaller PCC/rSDM (*Figure 5B, C*) and a diminishment of short distances between SNAP25 and syntaxin particles are observed (*Figure 5—figure supplement 2B*). In SNAP25 mobility measurements employing fluorescence recovery after photobleaching (FRAP), the same SNAP25 constructs showed that overexpression of syntaxin 1A slows down SNAP25 diffusion only if the N-terminal SNARE domain is present (*Halemani et al., 2010*). Hence, syntaxin 1A clusters associate with SNAP25 via the N-terminal SNARE domain which is noticed in a higher PCC/rSDM in super-resolution microscopy (*Figure 5B, C*) and a slower SNAP25 diffusion in FRAP (*Halemani et al., 2010*), respectively.

Setting aside the mesoscale organization and considering the nanoscale organization, what could the internal molecular anatomy of the syntaxin 1A/SNAP25 assemblies be? One possibility would be that only molecules at the border of the entities interact. In single-molecule localization experiments, the molecule concentration thins out towards the periphery, both in syntaxin 1A and SNAP25 entities, which is why syntaxin 1A–SNAP25 complexes could form in areas of lower packing around the core of syntaxin clusters (*Bar-On et al., 2012*). In a simplified view, in this scenario we deal with two syntaxin cluster phases, a pure cluster core and a cluster annulus containing heteromeric complexes. However, there may be no annulus. Instead, the syntaxin-cluster trunk could constitute a tight oligomeric bundle of alpha-helical SNARE-domains, serving as a binding surface for SNAP25. The SNARE domain of SNAP25 could zipper into one of the grooves of the trunk, becoming alpha-helical as well (*Figure 6*), similar to the SNARE-zippering reaction upon SNARE-complex assembly (*Brunger et al., 2019*; *Sutton et al., 1998*). This model is supported by the finding that in the above-mentioned FRAP measurement, SNAP25 mobility increases after introducing helix breakers into the N-terminal SNARE domain (*Halemani et al., 2010*).

An alternative scenario would be clustering of binary syntaxin 1A–SNAP25 complexes, in other words, there are no longer distinguishable syntaxin 1A and SNAP25 entities. This interpretation is challenged by the following two arguments. First, microscopic overlap between syntaxin clusters and SNAP25 crowds is only partial in the sense that they do not overlap concentrically (*Figure 4—figure supplement 4*; see also *Halemani et al., 2010*). In single-molecule imaging experiments, the majority of SNARE molecules are located in clusters and about half of the clustered syntaxin or SNAP25 molecules overlap with the nearest SNAP25 or syntaxin cluster, respectively (*Pertsinidis et al., 2013*). Regarding the high abundance of SNAP25 in PC12 cells, one may not expect half of the SNAP25 crowds in close proximity to syntaxin clusters. However, it should be noted that in Pertsinidis et al. mouse hippocampal neurons were used and that in a similar sample (rat synaptosomes) SNAP25 is only slightly more abundant than syntaxin 1A (*Wilhelm et al., 2014*). Moreover, as Pertsinidis et al. point out, not all molecules present may be detected, because steric hindrance does not allow binding of an antibody to every molecule in the cluster (*Pertsinidis et al., 2013*). Finally, the large fraction of SNAP25 being close to syntaxin clusters can also be explained by several SNAP25 crowds gathering around

one syntaxin cluster. In any case, analyzing overlap between two types of densely packed molecules is technically limited and provides no definite answer. For this reason, the following second argument is more conclusive. SNAP25 exceeds the concentration of syntaxin 1A, in the condition Stx-full roughly still about 5-fold (see above). If clusters would be large oligomers of syntaxin–SNAP25 complexes, in the condition Stx-full, 80 % of SNAP25 should not respond to syntaxin 1A overexpression. The strong effect we observe suggests that only a few molecules from the SNAP25 crowds directly interact with syntaxin clusters. Therefore, SNAP25 and syntaxin 1A entities may largely preserve their identities, interacting only at their peripheries. However, under certain conditions the excess of SNAP25 may be just too large to be controlled by syntaxin 1A. For instance, in the control of *Figure 4*, in which only SNAP25 is elevated, the cellular SNAP25:syntaxin 1A ratio is expected to increase up to 20- to 30-fold (the 10-fold physiological ratio doubles/triples using membrane sheets from the middle expression range, *Figure 1—figure supplement 3B*). Under this condition, the endogenous syntaxin 1A is no longer capable of re-organizing the majority of SNAP25, explaining why SNAP25 is spread all over the membrane, also in areas lacking any syntaxin 1A (*Figure 4A*, upper row).

## Possible impact of mesoscale organization on the SNARE-assembly pathway

In cell-free experiments, several types of syntaxin 1A/SNAP25 complexes form. An early report found that two syntaxin and one SNAP25 molecule assemble into a tight 2:1 ([syntaxin 1A]$_2$:SNAP25) complex considered to be an 'off-pathway' dead-end-complex actually inhibiting the fusion reaction (*Fasshauer and Margittai, 2004*). In liposome fusion assays, fusion rates increase when the 2:1 complex is avoided by using a small fragment of synaptobrevin for stabilizing a 1:1 complex, that acts as an acceptor of synaptobrevin from the liposome destined to fuse (*Pobbati et al., 2006*). The current view is that the 1:1 complex acts as an acceptor of synaptobrevin in vivo as well. This raises the question, how can unfavourable complexes in the plasma membrane be avoided? The law of mass action predicts that the more SNAP25 is available, the less (syntaxin 1A)$_2$:SNAP25 form. It has been previously suggested that more abundant SNAP25 disfavours the 2:1 complex and that differences in the clustering mechanisms between SNAP25 and syntaxin 1A increases the ratio between reactive SNAP25 and syntaxin 1A (*Halemani et al., 2010*). Our study undermines this theory. In accordance with others (*Knowles et al., 2010*) we find that in PC12 cells, aside from possible clustering effects, SNAP25 is manifold more abundant than syntaxin 1A. Hence, clustering in combination with the mesoscale organization could increase the abundancy of 1:1 'on-pathway' and minimize 2:1 'off-pathway' complexes, which could have a positive effect on the formation of fusion-relevant complexes.

## Molecular crowding in biological matter

'Like attracts like' is often observed in nature, from atoms to galaxies. With respect to cellular architectures, macromolecules tend to flock together beginning with the formation of dimers that may grow to larger oligomers. This localizes different species into different crowds, but in order to interact they have to get together. How this happens for SNAREs is revealed by our study. Compared to cell-free experiments in the absence of mesoscale organization, interacting crowds in the plasma membrane may preclude or favour specific reaction pathways. Hence, mesoscale organization of proteins is a cellular characteristics that could control protein assembly pathways in a crowded environment.

## Materials and methods

**Key resources table**

| Reagent type (species) or resource | Designation | Source or reference | Identifiers | Additional information |
|---|---|---|---|---|
| Cell line (*rattus norvegicus*) | PC12 cells | Other, *Weber et al., 2017* | | Gift from Rolf Heumann (Bochum, Germany), similar to clone 251; cell line maintained in lab |
| Cell line (*Homo sapiens*) | HepG2 cells | Cell line services | Cat# 300198; RRID:CVCL_0027 | Cell line maintained in lab |

*Continued on next page*

*Continued*

| Reagent type (species) or resource | Designation | Source or reference | Identifiers | Additional information |
|---|---|---|---|---|
| Strain, strain background (*Escherichia coli*) | BL21 (DEf3) | Stratagene (Agilent Technologies), *Parisotto et al., 2014* | | |
| Recombinant DNA reagent | Stx-full | *Merklinger et al., 2017* | N/A | Rat syntaxin 1A (NP_446240.2) in pEGFP-N1 vector (Clontech) |
| Recombinant DNA reagent | Stx-ΔS | *Merklinger et al., 2017* | N/A | Rat syntaxin 1 A deletion construct (aa 191–2818 missing) in pEGFP-N1 vector (Clontech) |
| Recombinant DNA reagent | GFP-SNAP25; SNAP25-WT | *Halemani et al., 2010* | N/A | Rat SNAP25B (NP_112253.1) in pEGFP-C1 vector (Clontech) |
| Recombinant DNA reagent | SNAP25-ΔN | *Halemani et al., 2010* | N/A | Rat SNAP25B deletion construct (aa 14–79 missing) in pEGFP-C1 vector (Clontech) |
| Recombinant DNA reagent | SNAP25-ΔC | *Halemani et al., 2010* | N/A | Rat SNAP25B deletion construct (aa 143–202 missing) in pEGFP-C1 vector (Clontech) |
| Recombinant DNA reagent | SNAP25-ΔN/C | *Halemani et al., 2010* | N/A | Rat SNAP25B deletion construct (aa 14–79 and 143–202 missing) in pEGFP-C1 vector (Clontech) |
| Peptide, recombinant protein | Munc18; Munc18-1 | *Parisotto et al., 2014* | N/A | cDNA of rat Munc18-1 subcloned into the pEG(KG) vector (gift of Dr. Richard Scheller) |
| Antibody | anti-HPC-1 (mouse monoclonal) | Sigma-Aldrich | cat# S0664; RRID:AB_477483 | IF: (1:500) WB: (1:1000) |
| Antibody | anti-SNAP25 (clone 71.1, mouse monoclonal) | Synaptic-Systems | cat# 111-011 | WB: (1:1000) |
| Antibody | anti-SNAP25 (rabbit polyclonal) | Synaptic Systems | cat# 111-002 | IF: (1:200) |
| Antibody | anti-$\alpha$-Tubulin (rabbit polyclonal) | Cell Signaling | cat# 2144 S | WB: (1:5000) |
| Antibody | anti-β-actin (clone 13E5, rabbit polyclonal) | Cell Signaling | cat# 4970 | WB: (1:5000) |
| Antibody | anti-GFP (rabbit polyclonal) | Chromotek | cat# PABG1; RRID:AB_2749857 | WB: (1:2000) |
| Antibody | anti-myc (clone Myc.A7, mouse monoclonal) | Invitrogen | cat# MA1-21316; RRID:AB_558473 | WB: (1:1000) |
| Antibody | 16L1-312F (mouse monoclonal) | Aliquot kindly provided by Luise Florin, University Medical Centre Mainz, *Knappe et al., 2007* | | IF: (1:500) |
| Antibody | anti-GFP nanobody (labelled with ATTO647) | Chromotek | cat# gba647n-100 | IF: (1:200) |
| Antibody | anti-GFP nanobody (labelled with ATTO594) | Chromotek | cat# gba594n-100 | IF: (1:200) |
| Commercial assay or kit | anti-GFP agarose beads | Chromotek | cat# gta-20 | (25 µl beads per probe) |
| Chemical compound, drug | isopropyl β-D-1-thiogalactopyranoside | Peqlab brand | cat# 730–1497 | (0.3 mM) |
| Software, algorithm | ImageJ | NIH; *Schindelin et al., 2012*, | RRID:SCR_003070 | |

*Continued on next page*

*Continued*

| Reagent type (species) or resource | Designation | Source or reference | Identifiers | Additional information |
|---|---|---|---|---|
| Software, algorithm | MatLab | MathWorks | RRID:SCR_001622 | |
| Software, algorithm | GraphPad Prism 6 | GraphPad software | RRID:SCR_002798 | |

## Plasmids

We used plasmids encoding for a C-terminally triple-myc tagged syntaxin 1A (Stx-full) from rat (NP_446240.2) and a triple myc-tagged variant where the N-terminal part of the SNARE domain (aa 191–218) is deleted, named Stx-ΔS (*Merklinger et al., 2017*). For GFP-SNAP25 expression, we used a plasmid encoding for a monomeric variant of EGFP fused N-terminally to rat SNAP25B (NP_112253.1). In SNAP25 deletion constructs the following sequences were deleted (amino acids are given in brackets): SNAP25-ΔN (14–79), SNAP25-ΔC (143–202), and SNAP25-ΔN/C (14–79 and 143–202) (*Halemani et al., 2010*). For bacterial expression of rat Munc18-1, the cDNA was subcloned into the pEG(KG) vector (a kind gift of Dr. Richard Scheller).

## Protein purification

The protein Munc18-1 (in the main text referred to as Munc18) was expressed and purified as described previously (*Parisotto et al., 2014*). Briefly, GST-Munc18-1 was expressed in *E. coli* BL21 (DEf3) cells after induction with 0.3 mM isopropyl β-D-1-thiogalactopyranoside (IPTG; Peqlab brand, cat# 730-1497) and 10 mM glucose at 16°C overnight. After lysis of the cells, insoluble material was removed by centrifugation at 35,000 rpm and 4°C for 1 hr. The supernatant was incubated with glutathione sepharose 4 fast flow (GE Healthcare Biosciences, cat# GE17-5132-01) and Munc18-1 was eluted from the beads by cleaving off the GST tag with thrombin (Merck, cat# 605195), which was subsequently inactivated with Pefabloc (final concentration: 2 mM; Roche, cat# 11429876001). Finally, the buffer of the protein was exchanged against 25 mM HEPES (4-(2-hydroxyethyl)-1-piperazineethanesulfonic acid), pH 7.4, 135 mM KCl, and 1 mM DTT at 4 °C using a PD10 column (GE Healtcare Biosciences, cat# 17085101).

## Cell culture and generation of membrane sheets

PC12 cells (*Weber et al., 2017*) were cultured at 5 % $CO_2$ in Dulbecco's modified Eagle medium (DMEM) with high (4.5 g/l) glucose (Gibco, Thermo Fisher, cat# 61965-026) supplemented with 10 % horse serum (BioWest, cat# S0900-500), 5 % fetal bovine serum (PAN Biotech, cat# P30-3021), and 100 U/ml penicillin + 100 ng/ml streptomycin (PAN Biotech, cat# P06-07100). Cells were tested negative for mycoplasma (Eurofins, barcode number 59189510). HepG2 cells (Cell Line Services, Eppelheim, Germany, cat# 300198; RRID:CVCL_0027) were cultured at 5 % $CO_2$ in MEM Eagle Medium (PAN Biotech, cat# P04-08509), supplemented with 10 % fetal bovine serum (PAN Biotech, cat# P30-3021) and (100 U/ml)/streptomycin (100 ng/ml) (PAN Biotech, cat# P06-07100).

In single and double transfections, cells were transfected with 10 µg plasmid DNA of each construct employing the Neon Transfection System (Thermo Fisher Scientific, cat# MPK5000) according to the manufacturer's manual. For PC12 cells a 1410 V/30 ms pulse was applied, for HepG2 cells a 1200 V/50 ms pulse. After electroporation cells were plated onto poly-L-lysine (PLL) (Sigma, cat# P-1524)-coated glass coverslips (25 mm diameter, Menzel Gläser, Braunschweig, Germany) or in 6-well culture dishes and used 16–24 h afterwards for experiments. For membrane sheet generation, cells were subjected to a brief 100 ms ultrasound pulse (employing a Bandelin Sonoplus GM2070 Sonifier) in ice-cold sonication buffer (120 mM potassium glutamate, 20 mM potassium acetate, 20 mM HEPES, 10 mM EGTA; pH 7.2).

## Co-immunoprecipitation

For immunoprecipitation, 24 hr after transfection HepG2 cells were scraped, suspended in ice-cold phosphate-buffered saline (PBS; 137 mM NaCl, 2.7 mM KCl, 10 mM $Na_2HPO_4$, 1.7 mM $KH_2PO4$, pH 7.4), and pelleted. Cells were lysed in ice-cold Tris buffer (150 mM NaCl, 0.5 mM EDTA, 10 mM Tris,

pH 7.5) containing 1 % Triton X-100. Immunoprecipitation was done according to the manufacturer's manual (anti-GFP Agarose beads, cat# gta-20, Chromotek). In brief, the cell lysate was incubated for 1 hr at 4 °C with anti-GFP nanobodies covalently bound to agarose beads. Beads were collected, washed, and heated for 10 min at 95 °C in 2× sodium dodecyl sulphate (SDS) buffer (according to protocol) to detach the precipitated material from the agarose beads, which were subsequently collected by centrifugation.

## Sodium dodecyl sulphate–polyacrylamide gel electrophoresis and Western blotting

Transfected cells seeded in 6-well plates were used 24 hr after transfection. As a control, cells not undergoing electroporation were used. Cells were washed once with ice-cold PBS and afterwards lysed by adding 4× Lämmli buffer (40% wt/vol glycerol, 300 mM Tris–HCl, 10% wt/vol SDS, 10 % β-mercaptoethanol, pH 6.8). Samples were heated for 10 min at 95 °C and subjected to SDS–polyacrylamide gel electrophoresis (SDS–PAGE) analysis. For SDS–PAGE, a 4 % stacking gel and a 10 % running gel were used. Western blotting was performed in ice-cold Towbin buffer (25 mM Tris, 192 mM glycine, 20% vol/vol MeOH, pH 8.3). Then, nitrocellulose membranes were washed in PBS and afterwards blocked for 1 hr with Odyssey Blocking Buffer (Li-Cor, cat# 924-70001). As primary antibodies, we used a monoclonal mouse antibody raised against the N-terminal domain of syntaxin 1 A (diluted 1:1000; clone HPC-1, cat# S0664, Sigma-Aldrich; RRID:AB_477483), a mouse monoclonal SNAP25 antibody raised against the N-terminus (amino acids 20–40, part of the first SNARE domain of SNAP25) of rat SNAP25B (diluted 1:1000; clone71.1, cat# 111 011, Synaptic Systems), and a rabbit polyclonal antibody against α-Tubulin (diluted 1:5000; cat# 2144, Cell Signaling), the latter serving as a loading control.

For the ratio analysis of endogenous SNARE proteins in *Figure 1—figure supplement 1*, a rabbit polyclonal antibody against β-actin (diluted 1:5000; clone 13E5, cat# 4970, Cell Signaling) was used for the loading control, together with the previously mentioned mouse monoclonal antibodies for SNAP25 and syntaxin 1A. The nitrocellulose membrane was then stripped for 15 min at RT with NewBlot Stripping buffer (cat# 928-40028, Li-Cor) and subsequently incubated with a rabbit polyclonal antibody raised against GFP (1:2000; cat# PABG1, Chromotek; RRID:AB_2749857).

Analyzing immunoprecipitation samples, for detection of GFP-SNAP25 a rabbit polyclonal antibody raised against GFP (see above) was used, and for syntaxin a monoclonal mouse antibody recognizing the myc-tag (diluted 1:1000; clone Myc.A7, cat# MA1-21316, Invitrogen; RRID:AB_558473).

As secondary antibodies, we used IRDye 800 CW labelled goat anti-mouse (cat# 925-32210) and IRDye 680RD labelled goat anti-rabbit (cat# 925-68071) from Li-Cor (Lincoln, NE; both diluted 1:10,000).

Primary and secondary antibodies were diluted in Odyssey Blocking buffer, supplemented with 0.1 % Tween-20 and incubated overnight at 4 °C or for 1 hr at RT, respectively. Between primary and secondary antibody incubation membranes were washed with PBS containing 0.1 % Tween-20 (PBS-T) and finally with PBS before imaging using a Li-Cor Odyssey Classic Imaging System. Integrated band intensities corrected for background were quantified using the software ImageJ (*Schindelin et al., 2012*).

## Immunostaining for microscopic analysis

For microscopy, cells were used 16–24 hr after transfection. Membrane sheets were generated and fixed with 4 % PFA for 30 min at RT. In Munc18 incubation experiments, 16 hr after transfection with GFP-SNAP25, membrane sheets were generated and either directly fixed or after incubation for 15 min at RT without or with 6 µM Munc18 in incubation buffer (135 mM KCl, 1 mM DTT, 25 mM HEPES, pH 7.4). PFA was quenched with 50 mM NH₄Cl in PBS for 20 min at RT, followed by blocking in 3 % bovine serum albumin (BSA) in PBS for 1 hr at RT.

For *Figure 2*, *Figure 2—figure supplement 1*, *Figures 4 and 5*, *Figure 5—figure supplement 3*, samples were incubated for 1 hr at RT with a primary antibody raised against syntaxin 1 A (HPC-1; see above) diluted 1:500 in PBS containing 1 % BSA (1 % BSA–PBS). After washing with 0.5 % BSA–PBS, the samples were incubated in 1 % BSA–PBS for 1 hr at RT with a secondary antibody (AlexaFluor594 labelled donkey anti-mouse, diluted 1:200; cat# A-21203, Invitrogen) and a nanobody (ATTO647N-labelled anti-GFP nanobody, diluted 1:200; cat# gba647n-100, Chromotek). Excessive secondary

antibody/nanobody was washed off with PBS. Samples were embedded in ProLong Gold Antifade Mountant (Thermo Fisher Scientific, cat# P10144) on microscopy slides, sealed and stored at 4 °C prior to imaging.

For analysis of GFP-SNAP25 targeting to the plasma membrane (*Figure 1—figure supplement 2*), cells were fixed as described for the membrane sheets, followed by permeabilization with 0.2 % Triton-X in PBS for 3 min, before employing the same immunostaining as well.

Switching spectral properties (*Figure 4—figure supplement 7*), syntaxin 1A was labelled with the previously mentioned HPC-1 primary antibody, but in this case visualized employing a goat anti-mouse secondary antibody coupled with ATTO647N (diluted 1:200; cat# 50185, Sigma). For GFP-SNAP25 visualization, a nanobody labelled with ATTO594 was used (diluted 1:200; cat# gba594-100, Chromotek).

To determine endogenous and overexpressed protein levels on membrane sheets (*Figure 1—figure supplement 3*), endogenous and overexpressed syntaxin 1A was labelled with HPC-1 in combination with a goat anti-mouse StarRed-coupled secondary antibody (diluted 1:200; cat# 2-0002-011-2, Abberior). Overexpressed Stx-full was labelled with a primary polyclonal rabbit anti-myc antibody (diluted 1:200; clone 71D10, cat# 2278, Cell Signaling) together with a donkey anti-rabbit AlexaFluor594 secondary antibody (1:200; cat# A-21207, Invitrogen). Endogenous and overexpressed SNAP25 was labelled with a primary rabbit antibody raised against the C-terminal part of SNAP25 (diluted 1:200; cat# 111002, Synaptic Systems) in combination with a goat anti-rabbit secondary antibody coupled to StarRed (diluted 1:200; cat# 2-0012-011-9, Abberior).

## Epifluorescence microscopy

Epifluorescence microscopy was performed on membrane sheets using an Olympus IX-81 inverted microscope equipped with an ImagEM C9100-13 16-bit EM CCD camera (Hamamatsu Photonics, Shizuoka, Japan), a MT20E illumination system (Olympus, Tokyo, Japan), an Apochromat NA 1.49 ×60 oil objective (yielding in combination with the camera a pixel size of 266.67 nm), and filter sets for GFP (F36-525 EGFP) and Alexa Fluor 647 (F46-009 Cy5 ET) (both from AHF Analysentechnik). With these settings, we recorded images containing several membrane sheets.

## Confocal and STED microscopy

Membrane sheets were imaged employing a four-channel super-resolution STED microscope (Abberior Instruments, Goettingen, Germany; available at the LIMES institute imaging facility, Bonn, Germany) module in combination with an Olympus IX83 confocal microscope (Olympus, Tokyo, Japan), equipped with an UPlanSApo ×100 (1.4 NA) objective (Olympus, Tokyo, Japan). AlexaFluor594/ATTO594 was excited with a 561 nm laser and emission recorded with a 580–630 nm filter. ATTO647N/StarRed was excited with a 640 nm laser and detected with a 650–720 nm filter. The pinhole size was set to 1 AU. A pulsed STED laser 775 nm was used for depletion. STED images were recorded via time-gated detection with 0.96 ns delay and 8 ns gate width.

For *Figure 1—figure supplement 2* and *Figure 2—figure supplement 1*, images were recorded in the confocal mode. GFP was excited with an Argon 485 nm laser and emission was collected from 500 to 520 nm. Pixel size in all recordings was set to 25 nm.

To determine the point spread function at the specific laser power settings employed for STED microscopy, we used papillomavirus pseudovirions (PsVs, *Finke et al., 2020*) for calibration. PsVs (2 µl, which corresponds to ~4 × $10^7$ viral genome equivalents) were diluted in 1 ml DMEM (see above) and vortexed vigorously. 500 µl of this solution was pipetted onto a PLL-coated coverslip and PsVs were allowed to adhere overnight at 4 °C. The staining was done as described before for membrane sheets. As a primary antibody, the mouse monoclonal 16L1-312F antibody (*Knappe et al., 2007*) (diluted 1:500, aliquot kindly provided by Luise Florin [University Medical Center Mainz, Germany]), was used. Secondary antibodies were coupled with the dyes AlexaFluor594 (donkey anti-mouse, diluted 1:200; cat# A-21203, Invitrogen) and ATTO647N (goat anti-mouse, diluted 1:200; cat# 50185, Sigma-Aldrich).

## Image analysis

In confocal sections imaging the equatorial cell plane we determined the fraction of GFP-SNAP25 localizing to the plasma membrane. To this end, we manually drew three regions of interest (ROIs) in ImageJ outlining the outer border of the cell ('outer rim'), the cytoplasm ('inner rim'), and the nucleus

('nucleus') (for example see *Figure 1—figure supplement 2*). After background correction, from each ROI's mean intensity and size the integrated density was calculated. Then, we define 'outer rim' – 'inner rim' = 'plasma membrane signal', 'inner rim' – 'nucleus' = 'cytoplasm signal', and consequently 'plasma membrane fraction' = 'plasma membrane signal'/('cytoplasm signal' + 'plasma membrane signal').

In epifluorescence micrographs, the mean intensities per membrane sheet were analyzed in ImageJ using the 'binary mask' function. In brief, images were smoothed with a Gaussian blur ($\sigma = 3$) to better define the outline of the membrane sheets. From the smoothed image a binary mask was created for overlay with the raw image. The mean intensity of each membrane sheet (as defined by the binary mask) was measured in both channels. From the mean intensity values the background was subtracted, which was determined by measuring the mean intensity in a ROI placed on an area without membrane sheet.

STED micrographs were analyzed with the program ImageJ using a costum written macro (*Merklinger et al., 2017*). ROIs were placed in the SNAP25 channel and propagated to the syntaxin 1A channel. For spot analysis (size and maxima density), the macro uses the ImageJ function 'Find Maxima' to detect maxima. For SNAP25 noise reduction was necessary. To this end images were smoothed with a Gaussian blur ($\sigma = 0.5$) prior to analysis to reduce pixel noise and thereby improve maxima detection. The coordinates of detected maxima were used for maxima size analysis. The maxima size was determined by applying a vertical and a horizontal $31 \times 3$ pixel linescan at the maxima position. A Gaussian function was fitted to the intensity distribution. The full width at half maximum of the Gaussian was taken as maxima size. Depending on the best fit quality, the size value was taken either from the horizontal or vertical linescan. Maxima with a fit quality of $R^2 < 0.8$ and a non-centred peak (not in the middle third of the linescan) were excluded. For each individual membrane sheet, all size values were averaged. The size of PsV maxima was determined following the same procedure.

The maxima density was obtained by relating the number of maxima to the size of the analyzed area.

For 'ROI mean intensity' measurements, mean intensity values within the ROIs were manually measured using the function 'measure mean intensity', subtracting a background value. For SNAP25, we calculated the relative standard deviation of the mean (rSDM), which is a parameter describing the degree of clustering (previously described in *Zilly et al., 2011*). To this end, the standard deviation of the mean intensity was determined and related to the mean intensity. Box plots were created using the program GraphPad Prism.

The PCC describes the linear relationship between two variables and was calculated using ImageJ. The same ROIs used for the rSDM were also used to measure the PCC between two channels. As a co-localization control, to mimic a non-related distribution between the two channels, one channel was flipped vertically and horizontally before determining the control PCC ('flipped'). For epifluorescence microscopy, squared ROIs where placed on the plasma membrane sheets only if they covered a reasonably large fraction of the plasma membrane area. Due to the low magnification in the recordings and the irregular shape of the membrane sheets, much less data points were obtained in comparison to the measurement of the average intensity using a binary mask (see above).

The pair distribution functions, radial maxima statistics, and distances to nearest neighbours were computed from the list of maxima positions (coordinates of SNAP25 and syntaxin 1A maxima extracted from the custom written ImageJ macro which was used for STED micrograph analysis) using MATLAB. We computed distances between maxima of similar and different types. As references, we consider ideal gases with the same maxima densities. As in ideal gases the locations of particles are uniformly randomly distributed, a comparison provides information about the structure of distributions, for example, the enrichment of short distances. For the computation of the different statistics only particles at sufficient distance from the slide boundaries were considered.

## Acknowledgements

This work was funded by the Deutsche Forschungsgemeinschaft (DFG, German Research Foundation) – Project Number 112927078 – TRR 83.

## Additional information

### Funding

| Funder | Grant reference number | Author |
|---|---|---|
| Deutsche Forschungsgemeinschaft | Project Number 112927078 | Thorsten Lang |

The funders had no role in study design, data collection, and interpretation, or the decision to submit the work for publication.

### Author contributions
Jasmin Mertins, Conceptualization, Data curation, Formal analysis, Investigation, Validation, Visualization, Writing – original draft; Jérôme Finke, Data curation, Formal analysis, Investigation, Validation, Writing – original draft; Ricarda Sies, Data curation, Investigation; Kerstin M Rink, Resources, Writing – original draft; Jan Hasenauer, Formal analysis, Software; Thorsten Lang, Conceptualization, Funding acquisition, Project administration, Supervision, Writing – original draft

### Author ORCIDs
Jasmin Mertins  http://orcid.org/0000-0001-5182-5705
Thorsten Lang  http://orcid.org/0000-0002-9128-0137

### Decision letter and Author response
Decision letter https://doi.org/10.7554/eLife.69236.sa1
Author response https://doi.org/10.7554/eLife.69236.sa2

## Additional files

### Supplementary files
- Transparent reporting form
- Source data 1. Statistical reporting.

### Data availability
All data generated or analysed during this study are included in the manuscript and supporting files.

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
