## [Editor Report]

In this study, the effect of syntaxin on co-localization of SNAP-25 in plasma membranes of PC12 cells is studied. The degree of co-localization increases as the syntaxin concentration is increased (relative to that of SNAP-25), and this increase depends on the two SNARE domains of SNAP-25, in particular on the N-terminal domain. When Munc18 is added to the plasma membrane sheets, little effect is observed on the syntaxin and SNAP-25 co-localization.

---

## [Decision Letter]

**Decision letter after peer review:**

Thank you for submitting your article "The mesoscale organization of syntaxin 1A and SNAP25 is determined by SNARE-SNARE-interactions" for consideration by *eLife*. Your article has been reviewed by 3 peer reviewers, one of whom is a member of our Board of Reviewing Editors, and the evaluation has been overseen by Olga Boudker as the Senior Editor. The reviewers have opted to remain anonymous.

In this study, the effect of syntaxin on co-localization of SNAP-25 in plasma membranes of PC12 cells is studied. The degree of co-localization increases as the syntaxin concentration is increased (relative to that of SNAP-25), and this increase depends on the two SNARE domains of SNAP-25, in particular on the N-terminal domain. When Munc18 is added to the plasma membrane sheets, little effect is observed on the syntaxin and SNAP-25 co-localization.

Essential revisions:

1. Among the new findings is an increase in the Pearson Correlation Coefficient (PCC) when syntaxin is over-expressed (Figure 4A). It is surprising that PCC increases even further when syntaxin-deltaS is over-expressed, consider that the interaction between syntaxin-deltaS and SNAP-25 is reduced (Figure 3). The authors attribute this to an increase in syntaxin clustering without a change in SNAP-25 distribution. However, if so, the radial pair distribution function should be similar at shorter distances. Please calculate pair distribution functions for all images shown in Figures 4, 5, and 6.

2. Figure 6 suggests that addition of Munc18 does not change the distribution of SNAP-25 and syntaxin. As mentioned above, please calculate the pair distribution functions for SNAP-25 and syntaxin. More importantly, however, the apparent lack of an effect of Munc18 is perhaps not so surprising. First, Pertsinidis et al., (2013) found Munc18 near SNAP-25 and syntaxin. Thus, addition of Munc18 is not expected to alter the distribution of SNAP-25 and syntaxin unless Munc18 was knocked out in the PC12 cultures. Second, if SNAP-25 and syntaxin were to form complexes in the membrane sheets, they have to be disassembled first by the action of NSF/SNAP (Ma et al., Science 339, 421-425, 2013; Lai et al., Neuron 95, 591-607 2017; Choi et al., *eLife* 7, e36497, 2018). Thus, ideally, to make this experiment interesting, endogenous Munc18 should be deleted, or, NSF/SNAP/ATP should be added to the membrane sheets. In the absence of such additional experiments, the conclusions drawn from this experiment need to be substantially toned down.

3. The authors have published several articles addressing the mobility of SNAREs in similar membranes. Did they (or colleagues in the literature) encounter also mobility changes in SNAP25, in relation to the expression of sytaxin 1?

4. Is the correlation coefficient between syntaxin-deltaS and SNAP25 significantly larger in Figure 4 than that for syntaxin-full length over-expression? If so, this would be counterintuitive.

5. Although the authors go to significant lengths to ensure at population averaged levels exogenous Syntaxin1A and SNAP-25 are only 1.5-fold and 2.5-fold overexpressed, many of the conclusions are based on single cell measurements that span a 20-40-fold range of expression levels. Since the conclusions are based on trend-lines and behaviors over this much wider dynamics range, which the authors did not calibrate, it is not clear if the conclusions apply to conditions where the system has been taken significantly out of its physiological dynamic range (e.g. from cells that vastly overexpress these proteins). Please calibrate the single-cell intensities, for quantifying degree of overexpression vs. the amount of endogenous protein, and try to analyze and see if conclusions can be drawn from a near-physiological dynamic range.

6. The methodological details are not clearly presented and some controls are missing. Specifically, which color was used for each protein, what was the STED raw resolution. This details are important to evaluate the conclusions (e.g., are SNAP25 fluorescent spots "diffuse" because of lower resolution)? What happens if two colors are swapped? Co-localization and resolution controls are missing. Please mention in figures or figure legends the fluorophores used for each imaging experiment. Please mention the raw resolution achieved by the two-color STED, for each color.

7. Please change the dyes in the two-color experiment and verify that the diffuse vs. tight clusters are not because of differences in the imaging performance (resolution, background) in the two difference color channels.

8. A limitation in relating the observations about mesoscale organizations into clusters and function of these proteins is that the experiments did not focus on the organization at active zones/fusion sites. Thus, the discussion about functional relevance is purely speculative at this point and should be toned down.

9. Lines 242-244 "The non-random distribution results from syntaxin-clusters being part of multi-protein assemblies that cover only half of the membrane area (Lillemeier et al., 2006; Saka et al., 2014), or in other words, syntaxin-clusters are excluded from half of the membrane area." Please quantify non-randomness of the syntaxin cluster distribution.

10. Lines 305-306 "for simplification, in Figure 5 we do not show the syntaxin channel". Please show the syntaxin-SNAP25 overlay.*Reviewer #1:*

The authors study here the distribution of two plasma membrane SNARE proteins, syntaxin 1 and SNAP25. The two proteins have been described more than two decades ago to form the SNARE complex that drives neuronal exocytosis, together with VAMP2 (synaptobrevin 2). At the same time, their distribution on cellular membranes has been investigated in multiple works by several groups, including the group of the corresponding author (Dr. Lang), and both proteins have been found to form clusters.

The current manuscript analyzes the relative distribution of the two proteins, based on imaging different mutated forms of both syntaxin 1 and SNAP25, both with conventional optics and super-resolution. They conclude that:

– Syntaxin 1 expression changes the packing of SNAP25 in the membrane (Figure 2).

– Syntaxin 1 expression modifies the localization of SNAP25 (Figure 4).

– The N-terminal domain of SNAP25 is required for such changes (Figure 5).

– The SNARE interactor Munc18 does not influence the organization of SNAP25.

Taking into account that SNAP25 is substantially more abundant than syntaxin 1 (work from the Almers group, Knowles et al., PNAS, 2010), this work implies that syntaxin 1 is an unusually strong regulator of the organization of SNAP25, and that this regulation requires SNARE domain interactions.

Overall, this results in the novel conclusion that the well-known SNARE domain interaction of the two proteins, which is the basis of exocytosis, is also important in driving the mesoscale organization of the proteins. I find this conclusion novel, especially as SNARE domain interactions have not been shown to carry substantial weight beyond the nanoscale.

*Reviewer #2:*

This work presents new data on a previously identified phenomenon, clustering of Syntaxin1A and SNAP25 on cell membranes. The authors perform a series of quantitative and super-resolution fluorescence imaging experiments, using a set of Syntaxin1A and SNAP25 deletion mutants that are over-expressed in PC12 cells. The authors discover that syntaxin1A overexpression reduces the overall membrane fluorescence signal from ectopically expressed GFP-SNAP25, both for the GFP tag, as well as an Atto647N-labelled GFP nanobody, likely reflecting increased self-quenching of GFP, and reduced accessibility of GFP to the nanobody. The magnitude of these effects is reduced when, instead of WT, a mutant Syntaxin1A lacking part of the SNARE domain (Stx-∆S) is overexpressed. This suggests that interactions via the Syntaxin1A SNARE domain change the organization and accessibility of SNAP25. Further using two-color super-resolution imaging, the authors discover that the mesocale organization of SNAP25 changes upon Syntaxin1A overexpression: SNAP25 forms mostly diffuse clusters that partially overlap with more tight Syntaxin1A clusters. Upon Syntaxin1A overexpression, SNAP25 becomes more tightly clustered around regions with Syntaxin1A clusters, an effect observed for WT Syntaxin1A but not for Stx-∆S overexpression. The authors investigate which SNAP25 domain is responsible for re-organization of SNAP25 upon WT Syntaxin1A overexpression. When simultaneously overexpressing WT Syntaxin1A and WT GFP-SNAP25 or GFP-SNAP25 lacking one or both of its N- or C-terminal SNARE domains, the authors observed that the effect of re-organization of GFP-SNAP25 around Syntaxin1A clusters can be mostly attributed to the N-terminal SNAP25 SNARE domain. Finally, incubation of "unroofed" PC12 cells with recombinant Munc18 protein has no effect on the mesocale organization of SNAP25 and Syntaxin1A, suggesting that the observed effects are mostly attributed to SNARE-mediated interactions. The authors attempt to discuss these observations in the context of how mesoscale protein organizations, clustering and molecular crowding might regulate biochemical reactions in the cell.

Strengths:

Particular strength of the manuscript is the use of various Syntaxin1A and SNAP25 mutants, and the extensive quantitative analysis of super-resolution images of clustered proteins.

Weaknesses:

Although the authors go to significant lengths to ensure at population averaged levels exogenous Syntaxin1A and SNAP-25 are only 1.5-fold and 2.5-fold overexpressed, many of the conclusions are based on single cell measurements that span a 20-40-fold range of expression levels. Since the conclusions are based on trend-lines and behaviors over this much wider dynamics range, which the authors did not calibrate, it is not clear if the conclusions apply to conditions where the system has been taken significantly out of its physiological dynamic range (e.g. from cells that vastly overexpress these proteins).

Another weakness of the manuscript in its current form is that methodological details are not clearly presented – for instance which color was used for each protein, what was the STED raw resolution etc. – so the basis for the conclusions cannot be evaluated (is SNAP25 diffuse because of lower resolution? What happens if two colors are swapped?) Co-localization and resolution controls are missing.

Finally, an important limitation in relating the observations about mesoscale organizations into clusters and function of these proteins is that authors did not look specifically at the organization at fusion sites. Thus the discussion about functional relevance appears speculative at this point.

[Editors’ note: further revisions were suggested prior to acceptance, as described below.]

Thank you for resubmitting your work entitled "The mesoscale organization of syntaxin 1A and SNAP25 is determined by SNARE-SNARE-interactions." for further consideration by *eLife*. Your revised article has been evaluated by Olga Boudker (Senior Editor) and a Reviewing Editor.

The manuscript has been improved but there are some relatively remaining issues that need to be addressed, as outlined below:

*Reviewer #1:*

I would like to thank the authors for addressing my concerns. I have no further comments.

*Reviewer #2:*

The authors replied to all of my comments, and I suggest that the manuscript can be published in its present form.

*Reviewer #3:*

The authors have adequately responded to most of the previous reviewers comments. The revised manuscript is significantly improved.

Recommendations.

Perhaps the two-color figure panels that use green and red combination could be changed to green-magenta, to facilitate readability.

Figure 2. Line 264-265 "For clarity, not the full 264 intensity range is shown that reaches values up to 25,000 a.u. and 40,000 a.u. for the ATTO647- and GFP-265 channel, respectively (in particular, the control includes also brighter membrane sheets)." I think the authors should show the complete set of data points used for the linear regression. If the figure becomes less clear, perhaps the full range can be plotted in a supplement figure panel.

Line 719 "'Like attracts like' is a fundamental principle in nature observed from atoms to galaxies", perhaps change to "'Like attracts like' is often observed in nature, from atoms to galaxies"

---

## [Author Response]

Essential revisions:1. Among the new findings is an increase in the Pearson Correlation Coefficient (PCC) when syntaxin is over-expressed (Figure 4A). It is surprising that PCC increases even further when syntaxin-deltaS is over-expressed, consider that the interaction between syntaxin-deltaS and SNAP-25 is reduced (Figure 3). The authors attribute this to an increase in syntaxin clustering without a change in SNAP-25 distribution. However, if so, the radial pair distribution function should be similar at shorter distances. Please calculate pair distribution functions for all images shown in Figures 4, 5, and 6.

We thank the referee for raising this important point. The referee is correct in assuming that the increase in shorter distances is similar for Stx-full and Stx-∆S (line 418, Figure 4 – Supplement 3A). For illustrating the substantial shortening of the distance between a SNAP25 crowd and its nearest syntaxin cluster we also calculated the probability density function (line 418, Figure 4 – Supplement 3B). The radial pair distribution function is included as well (line 418, Figure 4 – Supplement 3C).

We state on line 481:

“Overexpression of both Stx-full and Stx-∆S doubles the PCC between the syntaxin- and the SNAP25-channel (Figure 4A, see values stated in the overlay images). […] Smallest distances between maxima in the syntaxin- and SNAP25-channels are most frequent when SNAP25-WT is expressed, followed by SNAP25-ΔC, and SNAP25-ΔN/SNAP25-ΔN/C (Figure 5 —figure supplement 2).”

Regarding Figure 6, we have followed the suggestion of the referees to tone down this experiment, and moved Figure 6 to the supplement (now Figure 5 – Supplement 3).

We thank the referee again for this very interesting suggestion that helps explaining the unexpected similar increases in PCC (Figure 4) and strengthens the conclusions drawn from Figure 5.

2. Figure 6 suggests that addition of Munc18 does not change the distribution of SNAP-25 and syntaxin. As mentioned above, please calculate the pair distribution functions for SNAP-25 and syntaxin. More importantly, however, the apparent lack of an effect of Munc18 is perhaps not so surprising. First, Pertsinidis et al., (2013) found Munc18 near SNAP-25 and syntaxin. Thus, addition of Munc18 is not expected to alter the distribution of SNAP-25 and syntaxin unless Munc18 was knocked out in the PC12 cultures. Second, if SNAP-25 and syntaxin were to form complexes in the membrane sheets, they have to be disassembled first by the action of NSF/SNAP (Ma et al., Science 339, 421-425, 2013; Lai et al., Neuron 95, 591-607 2017; Choi et al., eLife 7, e36497, 2018). Thus, ideally, to make this experiment interesting, endogenous Munc18 should be deleted, or, NSF/SNAP/ATP should be added to the membrane sheets. In the absence of such additional experiments, the conclusions drawn from this experiment need to be substantially toned down.

We follow the latter suggestion of the referees and tone down the conclusions of the experiment, because after rethinking this issue, we agree that it is not expected that additional Munc18 alters the distribution of SNAP25 crowds. We have toned down the conclusions, shortened the paragraph in the Results section and removed the Munc18 paragraph from the discussion. We now state (line 570) “We also wondered whether the SNARE-domains are sufficient for mesoscale organization, or if perhaps Munc18 influences mesoscale organization by controlling the syntaxin 1A/SNAP25 interaction….Membrane sheets from cells expressing GFP-SNAP25 were incubated for 15 min with control buffer or 6 µM Munc18, fixed, stained and analyzed by STED-microscopy (Figure 5 —figure supplement 3)…addition of Munc18 has no significant effect on the rSDM and the maxima per µm^2^ and consequently appears to have no influence on the SNAP25 mesoscale distribution, although it may have one on the nanoscale-organization (e.g. the formation of complexes between single syntaxin 1A and Munc18 proteins).”

3. The authors have published several articles addressing the mobility of SNAREs in similar membranes. Did they (or colleagues in the literature) encounter also mobility changes in SNAP25, in relation to the expression of sytaxin 1?

Yes, we studied before mobility changes of SNAP25 in relation to syntaxin 1A expression. We cited the paper in the original version “We identified the N-terminal SNARE domain, that was a likely candidate, as a previous report showed it slows down SNAP25 mobility in a syntaxin dependent fashion (*Halemani et al., 2010*).” In the revised manuscript we expanded the discussion on line 643 “In SNAP25 mobility measurements employing fluorescence recovery after photobleaching (FRAP), the same SNAP25 constructs showed that overexpression of syntaxin 1A slows down SNAP25 diffusion only if the N-terminal SNARE domain is present (Halemani et al., 2010). Hence, syntaxin 1A clusters attach to SNAP25 via the N-terminal SNARE domain which is noticed in a higher PCC/rSDM in super-resolution microscopy (Figure 5B and C) and a slower SNAP25 diffusion in FRAP (Halemani et al., 2010), respectively.”

4. Is the correlation coefficient between syntaxin-deltaS and SNAP25 significantly larger in Figure 4 than that for syntaxin-full length over-expression? If so, this would be counterintuitive.

It is not significantly larger, in contrast to the differences between “control and Syx-full” and “control and Syx-ΔS”, that are highly significant. The statistical analysis of the differences between the PCCs is now provided in Figure 4 – Supplement 5 (line 440). Please see our reply to point #1 for an explanation why the PCC increases upon overexpression of Stx-ΔS as well.

5. Although the authors go to significant lengths to ensure at population averaged levels exogenous Syntaxin1A and SNAP-25 are only 1.5-fold and 2.5-fold overexpressed, many of the conclusions are based on single cell measurements that span a 20-40-fold range of expression levels. Since the conclusions are based on trend-lines and behaviors over this much wider dynamics range, which the authors did not calibrate, it is not clear if the conclusions apply to conditions where the system has been taken significantly out of its physiological dynamic range (e.g. from cells that vastly overexpress these proteins). Please calibrate the single-cell intensities, for quantifying degree of overexpression vs. the amount of endogenous protein, and try to analyze and see if conclusions can be drawn from a near-physiological dynamic range.

We fully agree. A critical issue is the excess of SNAP25 over syntaxin 1A. Knowles et al., (cited in the manuscript) find in PC12 cells a vast excess of 14-fold in the plasma membrane. Consequently, in our experiments under all conditions SNAP25 should be much more abundant than syntaxin 1A.

Because the issue is of great importance, we first verified the previously published SNAP25 excess in our PC12 cells by Western blot analysis. We found that SNAP25 is 10-fold more abundant than syntaxin 1A (Figure 1 – Supplement 1; line 164), which is very much in agreement with the above-mentioned 14-fold difference found in the literature. Regarding individual plasma membranes generated from overexpressing cells, we now show in Figure 1 – Supplement 3 (line 198) that syntaxin 1A and SNAP25 are elevated up to 9- and 4-fold, respectively, which assures a still 5-fold excess of SNAP25 over syntaxin 1A when using membrane sheets from the middle expression range. However, it is possible that the rSDM increases only at high expression levels, and not if SNAREs are close to physiological concentration. This has been addressed in Figure 4 – Supplement 6 (line 451) by plotting from 139 membrane sheets the rSDM against the syntaxin expression level (which varies by a factor of 10). We do not observe that the rSDM increases with expression level, rather the opposite is the case.

We state on line 348 “Finally, with reference to Figure 1 —figure supplement 3, the above-mentioned 5-fold increase in staining suggests that we imaged membrane sheets from the middle overexpression range, leaving still a 5-fold excess of SNAP25 over syntaxin 1A.” and on line 636 “We estimate that in our experiments the physiological SNAP25:syntaxin 1A ratio is roughly halved from 10-fold to 5-fold. However, the rSDM increase is not related to the diminishment of the physiological ratio, or an increase in SNARE concentration, because as shown in Figure 4 —figure supplement 6, we do not observe a trends towards larger rSDMs upon stronger overexpression.”

6. The methodological details are not clearly presented and some controls are missing. Specifically, which color was used for each protein, what was the STED raw resolution. This details are important to evaluate the conclusions (e.g., are SNAP25 fluorescent spots "diffuse" because of lower resolution)? What happens if two colors are swapped? Co-localization and resolution controls are missing. Please mention in figures or figure legends the fluorophores used for each imaging experiment. Please mention the raw resolution achieved by the two-color STED, for each color.

We apologize for not having presented clearly enough methodological details and certain controls.

a) As requested, we swapped colors (line 457, Figure 4 – Supplement 7; please note that AlexaFluor594 coupled GFP-nanobodies are not available which is why we used instead ATTO594 nanobodies; AlexaFluor594 and ATTO594 spectral properties are very close). We observe the same trends and fluorescent spot characteristics (see also reply to next point). We state in the figure legend (line 462) “As in Figure 4, Syntaxin 1A maxima are sharply defined and cover only a small area, whereas SNAP25 maxima appear more diffuse and are more widespread. (B and C) Compared to Figure 4, the same trends in ROI intensity, maxima number, maxima size and rSDM are also apparent after switching fluorophores.”

b) We have determined the raw resolution in the two channels (line 386, Figure 4 – Supplement 1) which is 67 nm and 66 nm for the AlexaFluor594- and ATTO647N-channel, respectively. We state on line 329“…using two-color super-resolution STED microscopy at ∼65 nm resolution (Figure 4 —figure supplement 1).”

c) We now mention in each figure legend the fluorophores used.

d) Co-localization controls for all PCC measurements are provided, added in Figure 2 suppl. 2 (line 285), Figure 4 —figure supplement 5 (line 440) and Figure 5 —figure supplement 1 (line 520). As control for two non-related images, the image of one channel is flipped horizontally and vertically and the PCC is calculated again. In all cases, the PCC in the flipped images is zero. Please note that in Figure 2 the original ROIs were not squares, a prerequisite for flipping. Therefore, the PCC analysis was repeated placing squared ROIs manually only on larger membrane sheets, which is the PCC analysis now includes less values. This change did not affect the statistics.

7. Please change the dyes in the two-color experiment and verify that the diffuse vs. tight clusters are not because of differences in the imaging performance (resolution, background) in the two difference color channels.

Please see our reply a to the above issue.

8. A limitation in relating the observations about mesoscale organizations into clusters and function of these proteins is that the experiments did not focus on the organization at active zones/fusion sites. Thus, the discussion about functional relevance is purely speculative at this point and should be toned down.

We have replaced ´Impact…´ by ´Possible impact…´ (line 690), eliminated ´an early step in the exocytic cascade´, and replaced in the abstract ´fusion sites´ by ´plasma membrane´ (line 34). In the discussion, now we just focus on how the ratio between SNAP25 and syntaxin 1A may influence the different types of SNAP25-syntaxin 1A complexes. We state on line 696 “The current view is that the 1:1 complex acts as an acceptor of synaptobrevin in vivo as well. This raises the question, how can unfavourable complexes in the plasma membrane be avoided? The law of mass action predicts that the more SNAP25 is available, the less (syntaxin 1A)2:SNAP25 complexes form. It has been previously suggested that more abundant SNAP25 disfavours the 2:1 complex and that differences in the clustering mechanisms between SNAP25 and syntaxin 1A increases the ratio between reactive SNAP25 and syntaxin 1A (Halemani et al., 2010). Our study undermines this theory. In accordance with others (Knowles et al., 2010) we find that in PC12 cells, aside from possible clustering effects, SNAP25 is manifold more abundant than syntaxin 1A. Hence, clustering in combination with the meso-scale organization could increase the abundancy of 1:1 “on-pathway” and minimize 2:1 “off-pathway” complexes, which could have a positive effect on the formation of fusion-relevant complexes.”

9. Lines 242-244 "The non-random distribution results from syntaxin-clusters being part of multi-protein assemblies that cover only half of the membrane area (Lillemeier et al., 2006; Saka et al., 2014), or in other words, syntaxin-clusters are excluded from half of the membrane area." Please quantify non-randomness of the syntaxin cluster distribution.

We compared the distance to the nearest syntaxin cluster to a simulated random distribution (Figure 4 – Supplement 2, line 403). We state (line 338): “The distance from one syntaxin 1A cluster to the nearest other syntaxin-clusters is different from the one of uniformly randomly distributed clusters behaving like particles in an ideal gases (Figure 4 —figure supplement 2). Compared to an ideal gas, the most likely distances are larger, peaking around 180 nm.”

10. Lines 305-306 "for simplification, in Figure 5 we do not show the syntaxin channel". Please show the syntaxin-SNAP25 overlay.

Has been added to Figure 5 (line 500).

[Editors’ note: further revisions were suggested prior to acceptance, as described below.]

Reviewer #3 (Recommendations for the authors):The authors have adequately responded to most of the previous reviewers comments. The revised manuscript is significantly improved.Recommendations.Perhaps the two-color figure panels that use green and red combination could be changed to green-magenta, to facilitate readability.

– Has been changed as requested.

Figure 2. Line 264-265 "For clarity, not the full 264 intensity range is shown that reaches values up to 25,000 a.u. and 40,000 a.u. for the ATTO647- and GFP-265 channel, respectively (in particular, the control includes also brighter membrane sheets)." I think the authors should show the complete set of data points used for the linear regression. If the figure becomes less clear, perhaps the full range can be plotted in a supplement figure panel.

– We now show the complete set of data points. The data points previously shown in green and red are now shown in green and magenta (see comment above).

Line 719 "'Like attracts like' is a fundamental principle in nature observed from atoms to galaxies", perhaps change to "'Like attracts like' is often observed in nature, from atoms to galaxies".

– Has been modified as suggested (now in line 703).

– Additionally, we corrected for a mistake in the formula in the legend of Figure 5 and rewrote a small paragraph in the discussion to improve clarity of the argument.